# VB-LoRA: Extreme Parameter Efficient Fine-Tuning with Vector Banks

**Yang Li**
Dept. of Computer Science
Georgia State University
Atlanta, GA 30303
yli93@student.gsu.edu

**Shaobo Han**
Optical Networking and Sensing
NEC Laboratories America
Princeton, NJ 08540
shaobo@nec-labs.com

**Shihao Ji**[*]
School of Computing
University of Connecticut
Storrs, CT 06269
shihao.ji@uconn.edu

## Abstract

As the adoption of large language models increases and the need for per-user or per-task model customization grows, the parameter-efficient fine-tuning (PEFT) methods, such as low-rank adaptation (LoRA) and its variants, incur substantial storage and transmission costs. To further reduce stored parameters, we introduce a "divide-and-share" paradigm that breaks the barriers of low-rank decomposition across matrix dimensions, modules, and layers by sharing parameters globally via a *vector bank*. As an instantiation of the paradigm to LoRA, our proposed VB-LoRA composites *all* the low-rank matrices of LoRA from a shared *vector bank* with a differentiable top-$k$ admixture module. VB-LoRA achieves extreme parameter efficiency while maintaining comparable or better performance compared to state-of-the-art PEFT methods. Extensive experiments demonstrate the effectiveness of VB-LoRA on natural language understanding, natural language generation, instruction tuning, and mathematical reasoning tasks. When fine-tuning the Llama2-13B model, VB-LoRA only uses 0.4% of LoRA's stored parameters, yet achieves superior results. Our source code is available at https://github.com/leo-yangli/VB-LoRA. This method has been merged into the Hugging Face PEFT package[2].

## 1 Introduction

Parameter-efficient fine-tuning (PEFT) casts a new paradigm that leverages strong prior knowledge built in foundation models and adapts them to a wide range of downstream tasks by updating a small amount of trainable parameters [He et al., 2021]. Compared to prefix/prompt tuning [Li and Liang, 2021, Lester et al., 2021] or in-context learning [Brown et al., 2020], fine-tuning a large-scale pre-trained model yields better domain specialization dictated by high-quality datasets [Brown et al., 2020, Liu et al., 2022, Zhao et al., 2023]. This process can be repeated to suit the needs of ever-changing deployment scenarios and personalizations. However, the sheer volume of parameter space across a multitude of instantiations [Sheng et al., 2023] poses challenges for storage, transmission, and computation, especially for low-resource hardware and consumer-grade networks [Borzunov et al., 2024].

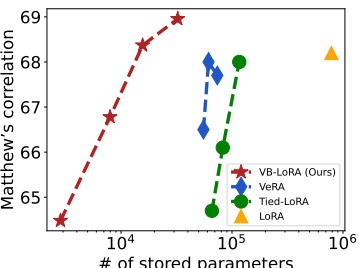

Figure 1: Comparison of the PEFT methods on RoBERTa-Large. Our VB-LoRA achieves higher scores with significantly smaller number of stored parameters.

To mitigate these challenges, various PEFT methods have been proposed by adding or adapting a small amount of trainable parameters per task without sacrificing performance [Houlsby et al., 2019,

---

[*]Part of the work was done while the author was affiliated with Georgia State University.
[2]https://huggingface.co/docs/peft/en/package_reference/vblora

38th Conference on Neural Information Processing Systems (NeurIPS 2024).

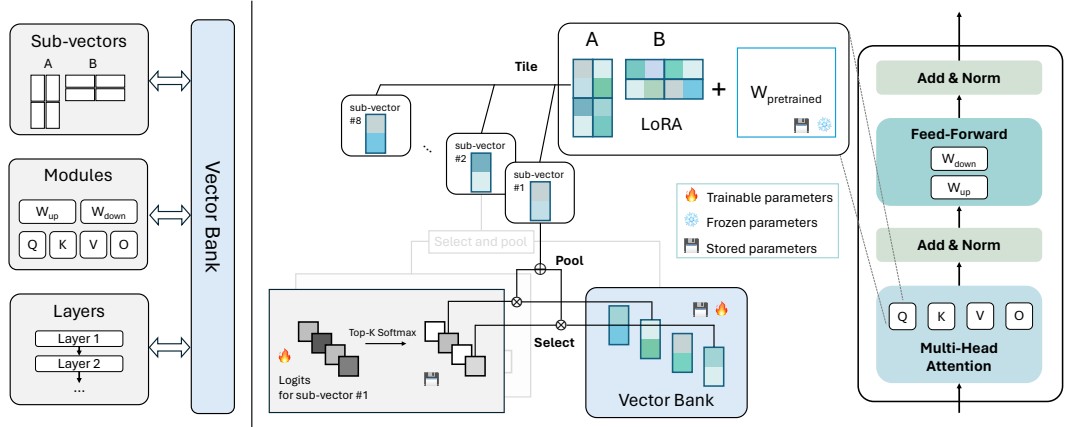

Figure 2: **Left**: The model parameters can be represented as a composition of vectors from a *vector bank*, which is shared across sub-vectors, modules and layers. **Right**: Architecture of VB-LoRA. We use a top-$k$ softmax function to select $k$ vectors from the vector bank. The selected vectors are then pooled into a sub-vector, which is arranged at a desired position, forming the parameters of LoRA.

Karimi Mahabadi et al., 2021, Ding et al., 2023]. These methods exploit the dependencies among model parameters to reduce the redundancy. For example, Hu et al. [2021] propose the low-rank adaptation (LoRA) to approximate the accumulated gradient update for self-attention modules, and induces the intra-matrix parameter coupling. Renduchintala et al. [2024] further study the options of allowing the inter-matrix parameter sharing via weight tying across all the layers. In both cases, the number of trainable parameters is reduced significantly. These two methods stand at the two extremes of spectrum in deciding the range of model components reuse (locally or across-layers) and designating which low-rank matrices needs to be shared and updated. However, as the model size increases and the demand for user-customized models across various services rises, the expense of storing and transmitting the customizations for each combination escalates and emerges as a critical issue. Hence, investigating PEFT methods with significantly smaller number of trainable parameters has attracted a flurry of research interests [Kopiczko et al., 2024, Renduchintala et al., 2024].

This paper introduces VB-LoRA, extreme parameter-efficient fine-tuning with *vector banks* based on a simple yet effective "divide-and-share" paradigm. We push the limits of LoRA parameter efficiency by breaking the two barriers of low-rank decomposition: (1) locally within each module and each layer, and (2) only across the two original matrix dimensions (without division; see Sec. 3.2 for details). We argue that the parameters across different modules and layers can be shared, and thus the redundancy in parameters can be further reduced. In addition, by partitioning rank-one component vectors into sub-vectors, we introduce "virtual" dimensions such that deep structure in the parameter space can be represented by a highly compressed matrix factorization.

VB-LoRA draws inspirations from previous line of work on quantized tensor networks [Oseledets, 2010, Cichocki, 2014] in breaking the constraint of physical dimension for extreme parameter compression. Specifically, VB-LoRA reparameterizes LoRA's low-rank adaptation by a rank-one decomposition and then divides the resulting vectors into sub-vectors of the same size. A *global sharing* mechanism is then learnt based on a sparse top-$k$ admixture module. The same sized sub-vectors allows parameters to be shared across modules and layers at the sub-vector level. Moreover, compared to the post-hoc matrix compression methods [Oseledets, 2010, Khoromskij, 2011], VB-LoRA is end-to-end differentiable, and therefore the fine-tuning process is aware of the compressed form, enabling task-oriented compression. Figure 1 illustrates the parameter efficiency of VB-LoRA as compared with state-of-the-art PEFT methods. Our contributions are summarized as follows:

1. We introduce a "divide-and-share" paradigm that breaks the barriers of low-rank decomposition across matrix dimensions, modules, and layers by sharing parameters *globally* via a vector bank.

2. We reparameterize LoRA's low-rank decomposition by a rank-one decomposition, and divide the resulting vectors further into sub-vectors of the same size, enabling extreme parameter efficiency at the sub-vector level.

3. We propose a sparse top-$k$ module based on the admixture model to learn a global sharing mechanism, making our framework end-to-end differentiable and compression-aware.

4. Our method achieves extreme parameter efficiency while maintaining comparable or better empirical performance compared to the state-of-the-art PEFT methods on natural language understanding, natural language generation, instruction tuning, and mathematical reasoning tasks.

## 2 Related Work

**Exploit Global Redundancy for Enhanced Parameter Efficiency** The parameters of deep neural networks (DNNs) can be naturally divided by layers, heads, or types (MHA or FFN). While LoRA [Hu et al., 2021] only exploits the *intra-matrix* dependency, Tied-LoRA [Renduchintala et al., 2024] employs a simple weight tying scheme on the low-rank matrices $A$ and $B$ across layers to reduce the *inter-matrix* redundancy. When $A$ and $B$ are randomly initialized, frozen, and shared across all layers, Tied-LoRA degenerates to VeRA [Kopiczko et al., 2024], which only requires two scaling vectors to be updated, leading to impressive parameter efficiency. A concurrent work, LoRA-XS [Bałazy et al., 2024], further improves the parameter efficiency of LoRA by introducing small trainable matrices between frozen LoRA projection matrices, which are initialized using Singular Value Decomposition (SVD) of the pretrained module weights. Our VB-LoRA pushes the limits of LoRA parameter efficiency by sharing parameters globally across modules and layers at the sub-vector level.

On the low-dimensional reparameterization, Aghajanyan et al. [2020] empirically show that there exists a low-dimensional reparameterization that is as effective for fine-tuning as the full parameter space. The actualization of the random projection is achieved through the Fastfood transform [Le et al., 2013] for large-scale pre-trained language models. To make it structure-aware, a set of layer-wise scaling parameters are included as part of the training parameters. Following this intuition, we study the lightweight fine-tuning within LoRA based on the customized reparameterization that arises from the rank-one matrix decomposition.

Moreover, tensor decomposition has been leveraged for PEFT in ViT models [Jie and Deng, 2023] based on classical formats, such as tensor-train or Tucker [Kolda and Bader, 2009]. We find that forcing multilinear decomposition across multiple modes results in a higher rank number, which is detrimental to the objective of parameter compression. An indirect comparison of VB-LoRA to Jie and Deng [2023] can be conducted by referring the compression rate to LoRA. From this perspective, our VB-LoRA can be viewed as a customized tensor format endowed with a convex geometry structure, which is enabled by the sparse top-$k$ admixture model we proposed.

Compared to the deep fusion approach [Mazzawi et al., 2024] where LLM parameters are split and initialized using pre-trained smaller networks under a designed network growth mechanism, our parameter division operates on the rank-one component vectors. Sub-vector division allows for similar extensions to leverage pre-trained vector bank initializations from smaller models and distributed training using model parallelism.

**Parameter Modeling based on Sparse Admixture Models** Admixture models have been widely used in population genetics [Pritchard et al., 2000], topic modeling [Reisinger et al., 2010, Inouye et al., 2014], and hyperspectral unmixing [Li and Bioucas-Dias, 2008, Fu et al., 2015] to extract archetypal (or endmember) components from observed data. The archetypal components can be relaxed to have mixed sign [Ding et al., 2008] with identifiability guarantees [Lin et al., 2015]. Conventionally, parameters estimation are conducted based on linear programming [Chan et al., 2009] or combinatorial algorithms [Arora et al., 2013]. However, an involved integer programming problem arises when incorporating an extra top-$k$ constraint into the mixing weights that is especially challenging for the large-scale language models. In this work, we propose learning archetypal vector banks not from observed data but from model parameters of LLMs. By modifying the sparse top-$k$ module [Shazeer et al., 2016] commonly used in Mixture-of-Expert models [Jiang et al., 2024], the mixing weights and vector banks are optimized by back-propagation under the objective of downstream fine-tuning tasks. The proposed top-$k$ admixture model is model-agnostic in the sense that it can be readily integrated into any neural network parameters or accumulated gradient updates.

# 3 Proposed Method

## 3.1 Preliminaries: Transformer Architecture and LoRA Adapters

The transformer architecture [Vaswani et al., 2017] consists of $L$ layers, each containing two types of blocks: Multi-Head Attention (MHA) and Feed-Forward Network (FFN). We denote the query, key, value, and output matrices of MHA at layer $\ell$ as $\boldsymbol{\mathcal{W}}_t^\ell = \{\boldsymbol{W}_t^i\}_{i=1}^{N_h}$, $t \in \{q, k, v, o\}$, where $\boldsymbol{W}_t^i \in \mathbb{R}^{d \times d}$, and $N_h$ is the number of heads. Given $\text{FFN}(\boldsymbol{x}) = \boldsymbol{W}_{\text{down}} \text{ReLU}(\boldsymbol{W}_{\text{up}} \boldsymbol{x})$ with $\boldsymbol{x} \in \mathbb{R}^d$, viewing FFN as a multi-head operation, we further divide $\boldsymbol{W}_{\text{up}} \in \mathbb{R}^{cd \times d}$ and $\boldsymbol{W}_{\text{down}} \in \mathbb{R}^{d \times cd}$ into $c$ matrices of size $d \times d$, denoted by $\boldsymbol{\mathcal{W}}_{\text{up}}^\ell = \{\boldsymbol{W}_{\text{up}}^{\ell,i}\}_{i=1}^c$ and $\boldsymbol{\mathcal{W}}_{\text{down}}^\ell = \{\boldsymbol{W}_{\text{down}}^{\ell,i}\}_{i=1}^c$. $c = 4$.

Given a pre-trained matrix $\boldsymbol{W}_0 \in \mathbb{R}^{m \times n}$, LoRA [Hu et al., 2021] constrains the weight increments $\Delta \boldsymbol{W}$ as a low-rank decomposition $\Delta \boldsymbol{W} = \boldsymbol{B} \boldsymbol{A}$, where $\boldsymbol{B} \in \mathbb{R}^{m \times r}$, $\boldsymbol{A} \in \mathbb{R}^{r \times n}$ are trainable parameters, with $r \ll \min(m, n)$. VeRA [Kopiczko et al., 2024] further limits the trainable parameters to two scaling vectors $b$ and $d$, which form the diagonal elements of two diagonal matrices $\Lambda_b$ and $\Lambda_d$. Hence, VeRA can be expressed as $\Delta \boldsymbol{W} = \Lambda_b \boldsymbol{B} \Lambda_d \boldsymbol{A}$, where $\boldsymbol{B}$ and $\boldsymbol{A}$ are randomly initialized, frozen and shared across layers.

Collectively, we denote the model parameters of transformer as $\boldsymbol{\Omega} = \{\{\boldsymbol{\mathcal{W}}_q^\ell, \boldsymbol{\mathcal{W}}_k^\ell, \boldsymbol{\mathcal{W}}_v^\ell, \boldsymbol{\mathcal{W}}_o^\ell\} \cup \{\boldsymbol{\mathcal{W}}_{\text{up}}^\ell, \boldsymbol{\mathcal{W}}_{\text{down}}^\ell\}\}_{\ell=1}^L \in \mathbb{R}^{12L \times d \times d}$. In the sequel, we propose a *global* reparameterization on the weight increments of $\boldsymbol{W} \in \boldsymbol{\Omega}$ based on the LoRA decomposition $\Delta \boldsymbol{W} = \boldsymbol{B} \boldsymbol{A}$. we will show how extreme parameter efficiency can be achieved by (1) parameter sharing across matrix dimensions of $\boldsymbol{A}$ and $\boldsymbol{B}$ based on a rank-one decomposition and sub-vector partitions (Sec. 3.2), and (2) across modules and layers regardless of the index or matrix type (Sec. 3.3).

## 3.2 Divide-and-Share: a New Paradigm for Parameter Sharing

The low rank decomposition of LoRA can be *equivalently* expressed in a rank-one form as follows:

$$\Delta \boldsymbol{W} = \boldsymbol{B} \boldsymbol{A} = \sum\nolimits_{k=1}^r \boldsymbol{b}_k \otimes \boldsymbol{a}_k = \sum\nolimits_{k=1}^r \otimes_{i=1}^2 \boldsymbol{v}_k^{(i)}, \quad \boldsymbol{v}_k^{(1)} = \boldsymbol{b}_k, \quad \boldsymbol{v}_k^{(2)} = \boldsymbol{a}_k, \qquad (3.1)$$

where $\otimes$ denotes the outer product operator and $\boldsymbol{v}_k^{(i)}$ is a vector of size $d_i$.

**Divide**   Based on the rank-one decomposition above, we further represent each component vector $\boldsymbol{v}_k^{(i)}$ as a concatenation of a set of sub-vectors,

$$\boldsymbol{v}_k^{(i)} = \text{concat}(\mathbf{u}_{k,1}^{(i)}, \mathbf{u}_{k,2}^{(i)}, \ldots, \mathbf{u}_{k,d_i'}^{(i)}), \quad \mathbf{u}_{k,j}^{(i)} \in \mathbb{R}^b, \quad j \in \{1, \ldots, d_i'\}, \qquad (3.2)$$

where $\{d_i\}_{i=1,2}$ represents the size of the matrix dimension of $\Delta \boldsymbol{W}$. In general, $\{d_i\}_{i=1,2}$ are not equal across $\boldsymbol{A}$ and $\boldsymbol{B}$, and we choose $b$ as a common factor of $d_i$ such that $d_i' = d_i/b$ and $d_i' \in \mathbb{Z}$.

**Share**   To facilitate parameter sharing across model dimensions, we assume each sub-vector $\boldsymbol{u}_{k,j}^{(i)}$ as a top-$k$ admixture of basic elements from vector bank $\mathcal{B} = \{\boldsymbol{\alpha}_1, \ldots, \boldsymbol{\alpha}_h\}$, where $\boldsymbol{\alpha}_i \in \mathbb{R}^b$ for $i \in \{1, \ldots, h\}$, and is defined as follows (with the subscripts omitted for clarity):

$$\boldsymbol{u} = \sum\nolimits_{s=1}^h w_s(\boldsymbol{\sigma}) \boldsymbol{\alpha}_s, \quad \mathbf{w}(\boldsymbol{\sigma}) = \text{Softmax}(\text{TopK}(\boldsymbol{\sigma}, k)), \qquad (3.3)$$

where $\text{TopK}(\boldsymbol{\sigma}, k)_i = \sigma_i$ if $\sigma_i$ is among the top-$k$ of $\boldsymbol{\sigma}$ and $\text{TopK}(\boldsymbol{\sigma}, k)_i = -\infty$ otherwise. For each sub-vector $\boldsymbol{u}$, we introduce logits $\boldsymbol{\sigma} \in \mathbb{R}^h$ as its learnable parameters. We call the model expressed in Eq. 3.3 as the *top-k admixture module* (TKAM), which is differentiable. This design enables the joint learning of vector bank $\mathcal{B}$ and logits $\boldsymbol{\sigma}$ in an end-to-end manner, which is amenable for model fine-tuning to the downstream tasks.

The TKAM module promotes sparsity by selecting $k$ vectors of the largest logits from the vector bank. By setting $k \ll h$, we restrict the sub-vector $\boldsymbol{u}$ to be sparse. That is, in each iteration, the updates to the vector bank remain locally dominated – with at most $k$ basis vectors $\boldsymbol{\alpha} \in \mathcal{B}$ affected by the backpropagation through $\boldsymbol{u}$ – in the hope that the learnt vectors can be more specialized and the knowledge encapsulated in the vector bank can be activated and updated sparsely.

**Noise-free Top-$k$ module** The Noisy Top-$k$ Gating module [Shazeer et al., 2016] has been widely used to replace the fully connected layers with the Mixture of Experts (MoE) layers in large language models [Jiang et al., 2024]. In contrast, we use Eq. 3.3 to learn the selective sharing scheme across the rank-one component vectors without changing the original model. Due to the decomposition, we find that the cumulative gradient parameter updates are more sensitive than the original model parameters during the training process. This may be related to the training instability issues observed in hypernetworks [Ortiz et al., 2024], where parameters are generated by another parameterized model as well. Therefore, keeping zero noise in the gating function can help make the learning more efficient and stable. An ablation study of different vector selection methods, including Gumbel-softmax, is provided in Sec. 4.5.

### 3.3 Breaking Boundaries of LoRA for Global Parameter Sharing

While LoRA only applies the low rank decomposition to each individual weight increment, the boundary can be broken by the *divide-and-share* scheme we proposed in Sec. 3.2. Our divide-and-share approach can be interpreted as hierarchical and constrained tensor decomposition, which facilitates efficient global parameter sharing that goes beyond LoRA's low-rank representation of matrices.

The divide operator was first introduced in Quantized Tensor Train (QTT) for super compression of large-scale matrices [Oseledets, 2010, Cichocki, 2014]. For example, dyadic division reshapes a vector of length $L = 2^p$ into a $p$-dimensional array which facilitates the efficient Tensor Train decomposition to be used. Our divide operator instead applies to the rank-one component vectors $\boldsymbol{v}_k^{(i)}$, and the resulting hierarchical tensorial representation of $\Delta \boldsymbol{W}$ can be viewed as a Canonical Polyadic Decomposition (CPD) [Kolda and Bader, 2009] with component vectors $\boldsymbol{v}_k^{(i)}$ folded into 2-dimensional arrays with sub-vectors $\boldsymbol{u}_{k,j}^{(i)}$ as columns. Each sub-vector $\boldsymbol{u_i}$ is composed from a *globally* shared vector bank $\mathcal{B}$ via TKAM, where $\mathbf{i} = [\mathbf{j}, \mathbf{v}]$ is a multi-index including physical indices $\mathbf{j}$, such as module, layer, head, and left/right decomposed matrix, and virtual indices $\mathbf{v}$ (created from vector partition).

The share operator (TKAM module) can be viewed as a factor model with simplex constraints on the mixing weight (e.g., $k = 2$, the sub-vector $\boldsymbol{u}$ lies on the edges of the simplex) and common factors stored in $\mathcal{B}$. Let $\boldsymbol{u} \in \mathbb{R}^b$ and $\boldsymbol{u} = \sum_{s=1}^h \boldsymbol{\alpha}_s w_s$, where $\boldsymbol{\alpha}_s$ is the $s$-th factor, and $\boldsymbol{w}$ is the factor score for the sub-vector $\boldsymbol{u}$. We consider the following options for $\boldsymbol{w}$: (1) Admixture (convex combination): $\boldsymbol{w} \in [0, 1]^h$ and $\sum_{s=1}^h w_s = 1$, which is commonly used in various communities. (2) Sparse Admixture (TKAM): $\boldsymbol{w} \in [0, 1]^h$ and $\sum_{s=1}^h w_s = 1$ with only $k \ll h$ non-zero elements allowed. It's worth mentioning that adding the multi-index information to the vector selection mechanism can make the TKAM model structure-aware, potentially yielding additional benefits. One possibility is to make the logits of vector selection conditional on the embeddings of the layer, module, and matrix type, which can be implemented through a hypernetwork [Mahabadi et al., 2021]. However, we leave this for future work.

In summary, LoRA provides a *local* low-rank factorization for each $d_1 \times d_2$ matrix $\Delta \boldsymbol{W}$ independently. In contrast, our VB-LoRA introduces a *global* low-rank factorization on a $b \times |\{\mathbf{i}\}|$ matrix composed of partitioned rank-one vectors, where $|\{\mathbf{i}\}|$ denotes the cardinality of the index set including both physical and virtual indices. As we will see below, this differentiation can better leverage the redundancy in the cumulative gradients, leading to extreme parameter efficiency.

Figure 2 overviews our method. The left section demonstrates the high-level idea of VB-LoRA: the vector bank is shared across sub-vectors, modules, and layers. The right section details its architecture. To form each sub-vector, we use a top-$k$ softmax function to select $k$ vectors from the vector bank, which are then pooled into a sub-vector. These sub-vectors are arranged in the desired positions, forming the parameters for LoRA with negligible computational overhead. Algorithm 1 provides the PyTorch-like pseudocode for VB-LoRA, which can be seamlessly integrated into the PyTorch framework.

### 3.4 Parameter Count

In full fine-tuning, the number of trainable parameters is equal to the model size, i.e., $LMd^2$, where $L$ is the number of layers, $M$ is the number of fine-tuned modules, and $d$ is hidden dimension.

**Algorithm 1** Pseudocode of VB-LoRA in a PyTorch-like style

```
# d: hidden dimension; b: length of sub-vectors; r: rank; h: size of vector bank
# k: number of selected vectors used in the top-k admixture module
# logits: Each linear layer has two trainable parameters: logits_A and logits_B.
#         Both parameters have a shape of (d/b)*r*h.
# vector_bank: The shared vector bank with a shape of h*b.
# x and W: input and the original weight.

def get_low_rank_matrix(logits, vector_bank, k):
    topk_logits, topk_indices = logits.topk(k, dim=-1)
    topk_weights = torch.softmax(topk_logits, dim=-1)
    matrix = (topk_weights * vector_bank[topk_indices]).sum(-2)
    return matrix

def VBLoRA_forward(x, vector_bank, logits_A, logits_B, k):
    r = logits_A.shape[1]
    A = get_low_rank_matrix(logits_A, vector_bank, k).transpose(0, 1).reshape(r, -1)
    B = get_low_rank_matrix(logits_B, vector_bank, k).transpose(1, 2).reshape(-1, r)
    # For memory efficiency, we avoid explictly computing \delta W = B @ A.
    return x @ W + (x @ B) @ A
```

LoRA reduces this number to $2LMdr$, while VeRA further reduces it to $LM(d + r)$. The trainable parameters of LoRA and VeRA are the same as the parameters they need to store.

In VB-LoRA, the trainable parameters consist of two parts: the parameters of the vector bank $\mathcal{B}$ and the parameters of logits $\boldsymbol{\sigma}$. However, at the end of training, the logit parameters can be discarded and only the $k$ selected indices and the top-$k$ admixture weights need to be stored. Therefore, the stored parameters can be represented by a triplet $\Theta = \{\mathcal{B}, \mathcal{I}, \mathcal{V}\}$, where $\mathcal{B} \in \mathbb{R}^{h \times b}$ is a vector bank containing $h$ vectors of $b$-dimensional, $\mathcal{I} \in \mathbb{R}^{2 \times L \times M \times r \times (d/b) \times k}$ is the top-$k$ indices of the vectors in $\mathcal{B}$ for all sub-vectors, and $\mathcal{V} \in \mathbb{R}^{2 \times L \times M \times r \times (d/b) \times (k-1)}$ is the top-$k$ admixture weights used to composite the sub-vectors from the bank. It is worth noting that the top-$k$ admixture weights have only $k - 1$ degrees of freedom since they must be summed to 1. Additionally, depending on the size of the vector bank $h$, the indices $\mathcal{I}$ can be efficiently stored as unsigned integers (e.g., uint8 when $h \leq 256$), and hence, we count the number of parameters as the float32-equivalent size for a fair comparison. When we use $k = 2$ and uint8 for indices, the number of stored parameters of VB-LoRA is $hb + 3LMr(d/b)$. Unlike LoRA and VeRA, the number of parameters in VB-LoRA does not increase linearly with the model size (determined by $L$ and $d$) or the number of fine-tuned modules, i.e., $M$. While the second term of VB-LoRA's parameters is a linear function of $LMd$, the coefficient is $3r/b$, which is typically very small. For example, in our experiments, the typical values are $r = 4$ and $b = 256$, leading to a coefficient of 0.04, whereas the coefficient is $2r$ for LoRA and 1 for VeRA. Most of the parameters in VB-LoRA reside within the shared vector bank, whose size does not increase linearly with the model size or number of fine-tuned modules.

# 4 Experiments

In this section, we conduct a comprehensive evaluation of our method through a series of experiments. We begin by comparing VB-LoRA to the state-of-the-art PEFT methods: LoRA, VeRA, and Tied-LoRA on the GLUE benchmark. Next, we extend our analysis to natural language generation tasks using GPT-2, instruction tuning tasks on the Llama2, as well as mathematical reasoning tasks on Mistral and Gemma models. All our experiments were conducted on a server equipped with 8 NVIDIA A100 GPUs. For reproducibility, we provide detailed hyperparameters and specifications of computing resources for each experiment in the appendix. The source code is available at `https://github.com/leo-yangli/VB-LoRA`.

## 4.1 Natural Language Understanding

We adopt the General Language Understanding Evaluation (GLUE) benchmark[3] [Wang et al., 2018] to assess the performance of VB-LoRA across various natural language understanding tasks, including

---

[3]`https://gluebenchmark.com/`

Table 1: Results with RoBERTa$_{base}$ and RoBERTa$_{large}$ on the GLUE benchmark. The best results in each group are shown in **bold**. We report Matthew's correlation for CoLA, Pearson correlation for STS-B, and accuracy for all other datasets. Results for LoRA$_{qv}$ and VeRA$_{qv}$ are sourced from their respective original papers, while the other results are based on our implementations. We report the median performance from 5 runs using different random seeds.

| | Method | # Params | SST-2 | MRPC | CoLA | QNLI | RTE | STS-B | Avg. |
|---|---|---|---|---|---|---|---|---|---|
| BASE | FT | 125M | 94.8 | 90.2 | 63.6 | 92.8 | 78.7 | 91.2 | 85.2 |
| | LoRA$_{qv}$ | 0.295M | $95.1_{\pm0.2}$ | $89.7_{\pm0.7}$ | $63.4_{\pm1.2}$ | $93.3_{\pm0.3}$ | $86.6_{\pm0.7}$ | $91.5_{\pm0.2}$ | 86.6 |
| | VeRA$_{qv}$ | 0.043M | $\mathbf{94.6}_{\pm0.1}$ | $\mathbf{89.5}_{\pm0.5}$ | $\mathbf{65.6}_{\pm0.8}$ | $91.8_{\pm0.2}$ | $78.7_{\pm0.7}$ | $90.7_{\pm0.2}$ | 85.2 |
| | Tied-LoRA$_{qv}$ | 0.043M | $94.4_{\pm0.5}$ | $88.5_{\pm1.0}$ | $61.9_{\pm1.6}$ | $92.0_{\pm0.1}$ | $76.2_{\pm1.0}$ | $89.8_{\pm0.3}$ | 83.8 |
| | VB-LoRA$_{qv}$ *(Ours)* | **0.023M** | $94.4_{\pm0.2}$ | $\mathbf{89.5}_{\pm0.5}$ | $63.3_{\pm0.7}$ | $\mathbf{92.2}_{\pm0.2}$ | $\mathbf{82.3}_{\pm1.3}$ | $\mathbf{90.8}_{\pm0.1}$ | **85.4** |
| | VeRA$_{all}$ | 0.157M | $\mathbf{95.1}_{\pm0.4}$ | $88.7_{\pm0.5}$ | $64.5_{\pm1.0}$ | $92.3_{\pm0.2}$ | $81.9_{\pm1.4}$ | $90.2_{\pm0.3}$ | 85.5 |
| | Tied-LoRA$_{all}$ | 0.109M | $94.7_{\pm0.2}$ | $88.5_{\pm0.8}$ | $\mathbf{64.7}_{\pm0.8}$ | $\mathbf{92.4}_{\pm0.1}$ | $76.5_{\pm1.3}$ | $90.3_{\pm0.1}$ | 84.5 |
| | VB-LoRA$_{all}$ *(Ours)* | **0.027M** | $95.0_{\pm0.2}$ | $\mathbf{89.7}_{\pm0.2}$ | $64.3_{\pm1.4}$ | $92.3_{\pm0.2}$ | $\mathbf{82.3}_{\pm0.9}$ | $90.7_{\pm0.2}$ | **85.7** |
| LARGE | LoRA$_{qv}$ | 0.786M | $96.2_{\pm0.5}$ | $90.2_{\pm1.0}$ | $68.2_{\pm1.9}$ | $94.8_{\pm0.3}$ | $85.2_{\pm1.1}$ | $92.3_{\pm0.5}$ | 87.8 |
| | VeRA$_{qv}$ | 0.061M | $\mathbf{96.1}_{\pm0.1}$ | $90.9_{\pm0.7}$ | $68.0_{\pm0.8}$ | $94.4_{\pm0.2}$ | $85.9_{\pm0.7}$ | $91.7_{\pm0.8}$ | 87.8 |
| | Tied-LoRA$_{qv}$ | 0.066M | $94.8_{\pm0.6}$ | $89.7_{\pm1.0}$ | $64.7_{\pm1.2}$ | $94.1_{\pm0.1}$ | $81.2_{\pm0.1}$ | $90.8_{\pm0.3}$ | 85.9 |
| | VB-LoRA$_{qv}$ *(Ours)* | **0.024M** | $\mathbf{96.1}_{\pm0.2}$ | $\mathbf{91.4}_{\pm0.6}$ | $\mathbf{68.3}_{\pm0.7}$ | $\mathbf{94.7}_{\pm0.5}$ | $\mathbf{86.6}_{\pm1.3}$ | $\mathbf{91.8}_{\pm0.1}$ | **88.2** |
| | VeRA$_{all}$ | 0.258M | $\mathbf{96.6}_{\pm0.5}$ | $90.9_{\pm0.8}$ | $68.5_{\pm1.4}$ | $\mathbf{94.4}_{\pm0.4}$ | $85.9_{\pm1.2}$ | $\mathbf{92.2}_{\pm0.2}$ | 88.1 |
| | Tied-LoRA$_{all}$ | 0.239M | $94.8_{\pm0.3}$ | $90.0_{\pm0.4}$ | $66.8_{\pm0.1}$ | $94.1_{\pm0.1}$ | $82.3_{\pm2.0}$ | $91.6_{\pm0.2}$ | 86.6 |
| | VB-LoRA$_{all}$ *(Ours)* | **0.033M** | $96.3_{\pm0.2}$ | $\mathbf{91.9}_{\pm0.9}$ | $\mathbf{69.3}_{\pm1.5}$ | $\mathbf{94.4}_{\pm0.2}$ | $\mathbf{87.4}_{\pm0.7}$ | $91.8_{\pm0.2}$ | **88.5** |

similarity, paraphrase, and inference tasks. Following Kopiczko et al. [2024], we focus on six tasks from GLUE: CoLA [Warstadt et al., 2019] (linguistic acceptability), SST-2 [Socher et al., 2013] (sentiment analysis), MRPC [Dolan and Brockett, 2005] (paraphrase detection), STS-B [Cer et al., 2017] (semantic textual similarity), QNLI [Rajpurkar et al., 2018] (inference), and RTE (inference).

Our experiments are performed with RoBERTa$_{base}$ and RoBERTa$_{large}$ [Liu et al., 2019]. While LoRA and VeRA only finetune the query and value modules, we explore two fine-tuning strategies: query and value only (VB-LoRA$_{qv}$), and all linear modules (VB-LoRA$_{all}$), including $W_q, W_k, W_v, W_o$, $W_{up}$, and $W_{down}$. We create a vector bank of 90 vectors of a length of 256, initialized with a uniform distribution $\mathcal{U}(-0.02, 0.02)$. The logits are initialized with a normal distribution $\mathcal{N}(0, 0.01)$. The learning rates for the vector bank and logit parameters are set to 0.001 and 0.01, respectively. We set the rank to 4 and $k = 2$ for all our experiments.

Table 1 reveals that VB-LoRA achieves competitive or superior performance compared to VeRA and Tied-LoRA, while being more parameter efficient. For example, when fine-tuning the query and value modules on the RoBERTa$_{large}$ model, our method reduces the stored parameters to less than 40% of those required by VeRA or Tied-LoRA, while outperforming them across all tasks. These results suggest that model performance depends not only on the quantity of trainable parameters but also on how they are composed.

Moreover, the results consistently indicate that fine-tuning all modules, beyond just the query and value modules, enhances performance for all the methods. However, LoRA, VeRA and Tied-LoRA requires 2–4 times of the parameters in this case because their parameter counts increase linearly with the number of fine-tuned modules. In contrast, our method uses only 37.5% additional parameters as we maintain the same vector bank size but add additional parameters for indices and top-$k$ weights. Thus, with only 12.8% of the parameters compared to VeRA$_{all}$ (4% compared to LoRA$_{qv}$), our method achieves the best average performance.

## 4.2 Natural Language Generation

For natural language generation experiments, we fine-tune the GPT-2 Medium and Large models [Radford et al., 2019] on the E2E dataset[4] [Novikova et al., 2017], which contains approximately 42,000 training examples, 4,600 validation examples, and 4,600 test examples from the restaurant domain. We use a vector bank of size 256 for GPT-2 Medium and 350 for GPT-2 Large. The vector length is set to 256 and the rank is set to 4 for both models. To achieve the best performance, we fine-tune all attention layers and FFN layers. As shown in Table 2, our approach achieves competitive performance compared to VeRA, while requiring about 20% less stored parameters for both models.

---

[4]Licensed under CC BY-SA 4.0. URL: `https://github.com/tuetschek/e2e-dataset`

Table 2: Results with GPT-2 Medium and GPT-2 Large on the E2E benchmark. The results for FT and LoRA are taken from Hu et al. [2021], and the results for VeRA are taken from Kopiczko et al. [2024]. We report the mean of 3 runs using different random seeds.

| | Method | # Params | BLEU | NIST | METEOR | ROUGE-L | CIDEr |
|---|---|---|---|---|---|---|---|
| MEDIUM | FT | 354.92M | 68.2 | 8.62 | 46.2 | 71.0 | 2.47 |
| | LoRA | 0.35M | 68.9 | 8.69 | 46.4 | 71.3 | 2.51 |
| | VeRA | 0.098M | **70.1** | **8.81** | **46.6** | **71.5** | 2.50 |
| | VB-LoRA *(Ours)* | **0.076M** | 70.0 | **8.81** | **46.6** | **71.5** | **2.52** |
| LARGE | FT | 774.03M | 68.5 | 8.78 | 46.0 | 69.9 | 2.45 |
| | LoRA | 0.77M | 70.1 | 8.80 | 46.7 | 71.9 | 2.52 |
| | VeRA | 0.17M | **70.3** | 8.85 | **46.9** | 71.6 | **2.54** |
| | VB-LoRA *(Ours)* | **0.13M** | **70.3** | **8.86** | 46.7 | **72.2** | **2.54** |

## 4.3 Instruction Tuning

Instruction tuning is a process of fine-tuning model with a set of instructions or prompts to enhance its performance on specific instructions [Ouyang et al., 2022]. We first experiment on a general instruction tuning dateset. We use the Cleaned Alpaca Dataset [5], which improves the data quality of the original Alpaca dataset [Taori et al., 2023]. We evaluate the fine-tuned models on the MT-Bench[6] [Zheng et al., 2024], which contains 80 multi-turn questions.

Following Kopiczko et al. [2024], we fine-tune the Llama2 model [Touvron et al., 2023] within the QLoRA [Dettmers et al., 2023] framework[7], which aims to reduce memory usage when fine-tuning large language models on a single GPU. We utilize the quantization strategy provided by QLoRA, including 4-bit NormalFloat for storage data, BFloat16 for computation parameters, double quantization and paged optimizers to train it on a single GPU. Our fine-tuned models generate responses to these questions, and subsequently, GPT-4 is employed to review and evaluate the generated answers, assigning a quantitative score on a scale of 10. Note that aligning with VeRA, we report the score of the first turn of the conversation. Following Kopiczko et al. [2024], we apply VB-LoRA to all linear layers except the top one. For Llama2 7B, we use a vector bank of 2,048 vectors, each with a length of 256, and the rank is set to 4, resulting in a total of 0.8M stored parameters. For Llama2 13B, we use the same-sized vector bank but increase the rank to 6, leading to 1.1M stored parameters. For all the experiments, we train for one epoch.

The results are reported in Table 3. Notably, we report two sets of LoRA results for each experiment: one from our implementation and the other from Kopiczko et al. [2024], due to a noticeable discrepancy between the scores. Since we closely follow the experimental settings of Kopiczko et al. [2024], we speculate that the difference is due to changes in the GPT-4 model over time. However, comparing the relative improvements of VeRA and VB-LoRA with their respective implementations of LoRA remains fair. VB-LoRA achieves higher scores than LoRA while using only 0.5% (Llama2 7B) and 0.4% (Llama2 13B) of the stored parameters. While VeRA can reach similar scores with their implementation of LoRA, it requires more than twice of parameters compared to VB-LoRA.

## 4.4 Mathematical Reasoning

To evaluate mathematical reasoning capabilities, we fine-tune the Mistral-7B-v0.1 and Gemma-7B models on the MetaMathQA[8] [Yu et al., 2023] dataset and test them on GSM8K[9] [Cobbe et al., 2021] and MATH[10] [Hendrycks et al., 2021] datasets. We compare our results with the concurrent work LoRA-XS [Bałazy et al., 2024], following its experimental configuration. The result is shown in Table 4. Our method outperforms all baselines on GSM8K, with Mistral-7B utilizing only 0.4% of

---

[5]The original and cleaned Alpaca datasets are licensed under CC BY-NC 4.0. URLs: `https://huggingface.co/datasets/tatsu-lab/alpaca`, `https://huggingface.co/datasets/yahma/alpaca-cleaned`

[6]Licensed under CC BY 4.0. URL: `https://huggingface.co/datasets/lmsys/mt_bench_human_judgments`

[7]`https://github.com/artidoro/qlora`

[8]Licensed under MIT. URL: `https://huggingface.co/datasets/meta-math/MetaMathQA`

[9]Licensed under MIT. URL: `https://huggingface.co/datasets/openai/gsm8k`

[10]Licensed under MIT. URL: `https://github.com/hendrycks/math/`

Table 3: Results with Llama2 on MT-Bench, scored by GPT-4 out of 10. LoRA$^\dagger$ and VeRA are sourced from Kopiczko et al. [2024]. LoRA$^\ddagger$ and VB-LoRA are from our implementations. The discrepancy between LoRA$^\dagger$ and LoRA$^\ddagger$ may be due to changes in the GPT-4 model over time.

| Model | Method | # Parameters | Score |
|-------|--------|-------------|-------|
| | w/o FT | - | 4.79 |
| LLAMA2 7B | LoRA$^\dagger$ | 159.9M | 5.19 |
| | VeRA | 1.6M | 5.08 |
| | LoRA$^\ddagger$ | 159.9M | 5.63 |
| | VB-LoRA *(Ours)* | **0.8M** | **5.71** |
| | w/o FT | - | 5.38 |
| LLAMA2 13B | LoRA$^\dagger$ | 250.3M | 5.77 |
| | VeRA | 2.4M | 5.93 |
| | LoRA$^\ddagger$ | 250.3M | 6.13 |
| | VB-LoRA *(Ours)* | **1.1M** | **6.31** |

Table 4: Results with Mistral-7B and Gemma-7B models on the GSM8K and MATH Benchmarks. Specifically, in VB-LoRA, we use a vector bank size of 2,048 with $b = 256$, set the rank to 4, and train with a batch size of 128 for 2 epochs. The warm-up ratio is 0.02, and training uses a cosine learning rate scheduler, with an initial learning rate of 0.001 for the vector bank and 0.01 for the logits. The baseline results are taken from Bałazy et al. [2024].

| Model | Method | # Parameters | GSM8K | MATH |
|-------|--------|-------------|-------|------|
| | Full-FT | 7242M | 67.02 | 18.60 |
| MISTRAL-7B | LoRA | 168M | 67.70 | **19.68** |
| | LoRA-XS | 0.92M | 68.01 | 17.86 |
| | VB-LoRA *(Ours)* | **0.65M** | **69.22** | 17.90 |
| | Full-FT | 8538M | 71.34 | 22.74 |
| GEMMA-7B | LoRA | 200M | 74.90 | **31.28** |
| | LoRA-XS | 0.80M | 74.22 | 27.62 |
| | VB-LoRA *(Ours)* | **0.67M** | **75.96** | 28.90 |

the parameters compared to LoRA, and Gemma-7B using just 0.3%. Compared with LoRA-XS, our method outperforms on both evaluation datasets while using 70% (Mistral-7B) and 83% (Gemma-7B) of LoRA-XS parameters.

## 4.5 Ablation Study

We conduct an ablation study to examine the impact of each individual component of VB-LoRA. The experiments are performed on RoBERTa-large, fine-tuning only the query and value modules.

**Vector Selection Methods** Besides the top-$k$ admixture module (abbreviated as Top-$k$ below), there exist several commonly used discrete optimization methods for vector selection, including Noisy Top-$k$ [Shazeer et al., 2016], Gumbel-Softmax (GS), and Straight-Through Gumbel-Softmax [Jang et al., 2017, Maddison et al., 2016]. For Top-$k$ and Noisy Top-$k$, we evaluate the impact of different $k$ to the performances on the CoLA dataset. For GS and Straight-Through GS, we set the temperature $\tau = 1/3$ during training and use Top-1 and Top-2 Softmax for inference. Additionally, we explore "Select All", a special case of Top-$k$ with $k$ equals to the vector bank size $h$. As shown in Table 5, Noisy Top-$k$, GS, and Straight-Through GS significantly underperform Top-$k$ and "Select All". We hypothesize that random noise injected by these methods likely disrupts the parameters of vector bank, leading to instability in the learning process.

We further investigate the impact of $k$ to the training dynamics and performance of VB-LoRA. As discussed in Sec. 3.4, the choice of $k$ affects not only the model's performance but also the number of parameters to be stored. Hence, a smaller $k$ is generally preferred for improved parameter efficiency. Table 5 shows that $k = 2$ yields the best result on CoLA, whereas $k = 1$ performs significantly worse. To explain this, we delve into the training dynamics of VB-LoRA. As shown in Figure 3 (a), when $k = 1$, the selected vectors remain largely unchanged during training. In contrast, when $k > 1$, the model actively explore the vector bank as illustrated in Figure 3 (b) and (c), i.e., different vectors are selected and updated actively during the training process. Additionally, we observed that this vector exploration primarily occurs in the early stages of training, with updates becoming progressively sparser in later stages, as shown in Figure 5 in the appendix. This suggests that the vectors become increasingly specialized for specific sub-vectors as training progresses.

**Sub-vector Length $b$** VB-LoRA introduces a new virtual dimension that divides the original dimensions of LoRA matrices into sub-vectors of length $b$. Note that $b$ must be a common factor of all hidden dimensions to ensure compatibility across the entire model. However, the optimal value of $b$ is task-specific and requires tuning as a hyperparameter. Theoretically, with a fixed vector bank budget, a larger $b$ reduces the number of vectors in the vector bank, potentially making each vector less specialized. On the other hand, a smaller $b$ increases the number of trainable parameters and complicates the vector selection process. As shown in Table 6, a moderate $b = 256$ yields the best performance on the CoLA task.

Table 5: Ablation study of different vector selection methods. S.: Softmax, GS: Gumbel-Softmax, ST-GS: Straight Through Gumbel-Softmax.

| Method | Training | Inference | CoLA |
|--------|----------|-----------|------|
| Select All | S. | S. | $67.5_{\pm 1.2}$ |
| Top-$k$ | Top 1 S. | Top 1 S. | $66.9_{\pm 0.5}$ |
| | Top 2 S. | Top 2 S. | $\mathbf{68.3}_{\pm 0.7}$ |
| | Top 3 S. | Top 3 S. | $68.1_{\pm 1.3}$ |
| | Top 6 S. | Top 6 S. | $67.1_{\pm 0.5}$ |
| Noisy Top-$k$ | Noisy Top 1 S. | Top 1 S. | $45.3_{\pm 2.2}$ |
| | Noisy Top 2 S. | Top 2 S. | $62.6_{\pm 0.2}$ |
| GS | GS ($\tau$=1/3) | Top 1 S. | $57.1_{\pm 0.6}$ |
| | GS ($\tau$=1/3) | Top 2 S. | $57.3_{\pm 1.6}$ |
| ST-GS | ST-GS ($\tau$=1/3) | Top 1 S. | $55.6_{\pm 1.6}$ |
| | ST-GS ($\tau$=1/3) | Top 2 S. | $54.7_{\pm 1.2}$ |

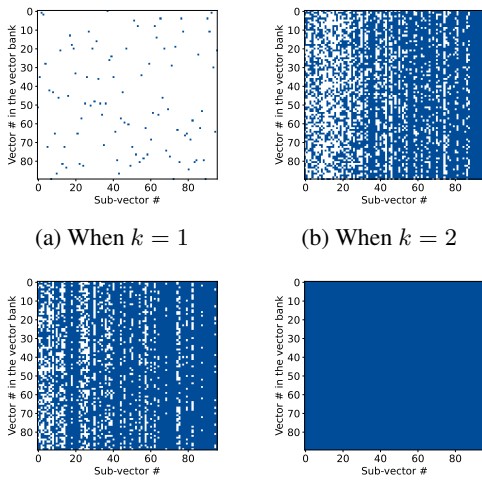

(a) When $k = 1$    (b) When $k = 2$

(c) When $k = 3$    (d) Noisy Top-2

Figure 3: VB-LoRA's vector selection footprints during training. The x-axis represents the 96 sub-vectors formed by the vectors from a bank of 90 vectors, while the y-axis represents the indices of selected vectors from the bank. The blue blocks indicate the selection footprint during training.

Table 6: Ablation study of sub-vector length.

| Length $b$ | Vector Bank Size | CoLA |
|------------|------------------|------|
| 128 | 240 | $67.0_{\pm 0.8}$ |
| 256 | 120 | $\mathbf{68.7}_{\pm 0.7}$ |
| 512 | 60 | $67.8_{\pm 0.8}$ |
| 1024 | 30 | $67.3_{\pm 1.1}$ |

## 5   Conclusion

This paper introduces a "divide-and-share" paradigm and a differentiable top-$k$ admixture module for extreme parameter-efficient fine-tuning with vector banks. Our proposed VB-LoRA achieves the competitive or higher accuracy while using significantly smaller number of stored parameters compared to the state-of-the-art PEFT methods, including LoRA, VeRA, and Tied-LoRA. In addition, VB-LoRA is model-agnostic and applicable to other PEFT methods [Ding et al., 2023], including inserted adapters [Karimi Mahabadi et al., 2021], prompt tuning [Qin et al., 2021], and BitFit [Ben Zaken et al., 2022]. Although VB-LoRA focuses on reducing the storage and transmission costs for LLM fine-tuning, we believe the proposed scheme can be extended to memory-efficient fine-tuning and parameter-efficient pre-training. We leave these for future exploration.

Fine-tuning a pre-trained model requires making design choices about which layers of the model should be frozen or updated. Multitask fine-tuning adds extra complexity about which parameters should be shared or task-specific. Along this line of work, Polytropon [Ponti et al., 2022] jointly learns a small inventory of LoRA adapters and a routing function that selects a variable-sized subset of adapters for few-shot adaptation. Caccia et al. [2023] emphasize the importance of routing granularity and further propose a finer-grained mixing across multiple heads. Following these works, it would be interesting to explore a finer-grained parameter transfer across tasks, heads, types, and layers at the sub-vector level for multitask fine-tuning.

**Limitations and broader impacts** Our experiments are limited to monomodal (text-based), monolingual (English), and LoRA-only settings. Additionally, our exploration of the vector bank is somewhat limited, as we only examine a small range of configurations for bank size and vector length. In terms of broader impacts, VB-LoRA reduces the storage and transmission costs of LLM adapters and demonstrates improved memory-efficiency, making customized LLMs more accessible. We do not foresee any negative societal impact beyond those generally associated with LLMs.

## Acknowledgments

We would like to thank the anonymous reviewers for their comments and suggestions, which helped improve the quality of this paper.

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

# A Appendix

## A.1 Hyperparameters and Computing Resources

The hyperparameters used for the natural language understanding, natural language generation and instruction tuning are provided in Table 7, 8 and 9. All experiments were conducted on a server equipped with 8 NVIDIA A100 80GB GPUs.

**Computation overhead** The proposed factorization in VB-LoRA is simple to implement in modern deep learning frameworks such as PyTorch, allowing us to fully leverage GPU acceleration. However, the use of subvector decomposition does introduce some computational overhead. This additional overhead is limited to the training phase and does not affect inference, as both LoRA and VB-LoRA merge their parameters back into the original model parameters during this stage.

**Memory efficiency** Despite the training time overhead, the reduced number of trainable parameters in VB-LoRA results in lower memory consumption. During LoRA fine-tuning, the forward pass is $z = Ax$, $H = Bz$, without the need to materialize $\Delta W$. This memory-saving technique can be seamlessly incorporated in VB-LoRA and has been implemented in our source code. Table 9 shows that VB-LoRA requires approximately 15%-20% more training time than LoRA, while it consumes less memory than LoRA in both the LLaMA2 7B model and LLaMA2 13B models.

Table 7: Hyperparameters and computing resources for natural language understanding experiments on the GLUE benchmark. Training time and GPU memory are reported as "query and value only" / "all linear modules". h: hour, m: minute.

| Model | Hyperparameter | SST-2 | MRPC | CoLA | QNLI | RTE | STS-B |
|---|---|---|---|---|---|---|---|
| | Optimizer | | | AdamW | | | |
| | Warmup Ratio | | | 0.06 | | | |
| | LR Schedule | | | Linear | | | |
| | Init. of the Vector Bank | | | $\mathcal{U}(-0.02, 0.02)$ | | | |
| | Init. of the Logits | | | $\mathcal{N}(0, 0.01)$ | | | |
| BASE | # GPUs | | | 1 | | | |
| | Epochs | 60 | 30 | 80 | 25 | 160 | 80 |
| | Learning Rate (Head) | 4E-3 | 4E-3 | 2E-2 | 1E-2 | 2E-2 | 2E-2 |
| | Learning Rate (Logits) | | | 1E-2 | | | |
| | Learning Rate (Vector Bank) | | | 1E-3 | | | |
| | Vector Bank Size | | | 90 | | | |
| | Vector Length | | | 256 | | | |
| | Rank | | | 4 | | | |
| | Max Seq. Len. | | | 512 | | | |
| | Batch Size Per GPU | | | 32 | | | |
| | Training Time | 8h / 10h | 27m / 40m | 80m / 100m | 5h / 6.5h | 50m / 1h | 1h / 80m |
| | GPU Memory | | | 24,552 MiB / 28,120 MiB | | | |
| LARGE | # GPUs | | | 1 | | | |
| | Epochs | 20 | 40 | 40 | 20 | 40 | 40 |
| | Learning Rate (Head) | 3E-3 | 3E-3 | 3E-3 | 2E-3 | 2E-3 | 6E-3 |
| | Learning Rate (Logits) | | | 1E-2 | | | |
| | Learning Rate (Vector Bank) | | | 1E-3 | | | |
| | Vector Bank Size | | | 90 | | | |
| | Vector Length | | | 256 | | | |
| | Rank | | | 4 | | | |
| | Max Seq. Len. | | | 128 | | | |
| | Batch Size Per GPU | | | 32 | | | |
| | Training Time | 2h / 3h | 12m / 20m | 30m / 45m | 3h / 4.5h | 10m / 15m | 20m / 30m |
| | GPU Memory | | | 9,804 MiB / 12,170 MiB | | | |

## A.2 Visualization of the Vector Selection

For visualization, we conducted experiments on the CoLA dataset using a 24-layer RoBERTa-large model with a vector bank of 30 vectors. We fine-tuned the query and value modules, setting the rank to 2 and the vector length to 1024, resulting in 192 sub-vectors.

Table 8: Hyperparameters and computing resources on natural language generation experiments on the E2E dataset. Training time and GPU memory are reported as "query and value only" / "all linear modules". h: hour, m: minute.

| Hyperparameter | Medium | Large |
|---|---|---|
| # GPUs | 1 | |
| Optimizer | AdamW | |
| Learning Rate Schedule | Linear | |
| Weight Decay | 0.01 | |
| Batch Size | 8 | |
| Epochs | 5 | |
| Warmup Steps | 500 | |
| Label Smooth | 0.1 | |
| Rank | 4 | |
| Vector Length | 256 | |
| Vector Bank Size | 256 | 350 |
| Learning Rate (Vector Bank) | 1E-3 | 1E-3 |
| Learning Rate (Logits) | 1E-2 | 1E-2 |
| Training Time | 3h | 3h |
| GPU Memory | 29,061 MiB | 29,282 MiB |

Table 9: Hyperparameters and computing resources on instruction tuning on the Cleaned Alpaca Dataset. h: hour. 7B: Llama2 7B, 13B: Llama2 13B.

| Hyperparameter | LoRA, 7B | LoRA, 13B | VB-LoRA, 7B | VB-LoRA, 13B |
|---|---|---|---|---|
| # GPUs | | 1 | | |
| Optimizer | | AdamW | | |
| Warmup Ratio | | 0.1 | | |
| Batch Size | | 4 | | |
| Accumulation Steps | | 4 | | |
| Epochs | | 1 | | |
| LR Schedule | | Linear | | |
| Vector Length | N/A | N/A | 256 | 256 |
| Rank | 64 | 64 | 4 | 6 |
| Vector Bank Size | N/A | N/A | 2048 | 2048 |
| Learning Rate (Vector bank) | N/A | N/A | 1E-3 | 1E-3 |
| Learning Rate (Logits) | N/A | N/A | 1E-2 | 1E-2 |
| Learning Rate (LoRA) | 4e-4 | 4e-4 | N/A | N/A |
| Training Time | 2h | 2.6h | 2.5h | 3h |
| GPU Memory | 8,467 MiB | 11,624 MiB | 6,872 MiB | 11,486 MiB |

Figure 4 displays the vectors selected by sub-vectors at the initialization (red) and at the end of training (blue), respectively. As we can see, most of the final selections differ from the initial selections, demonstrating the training dynamics of the vector selection process.

In Figure 5, we plot the footprint at different training periods. This visualization demonstrates that vector exploration predominantly occurs in the early stages of training, and the updates become progressively sparser in the later stages of training.

Figure 6 illustrates the sum of the top-k weights for each vector, grouped by the first, middle, and last 8 layers. It shows that certain vectors are favored by deeper layers, such as vectors #1 and #29, while some are favored by shallower layers, such as vectors #20 and #26.

We then group the same data with respect to query and value modules, as well as matrices A and B, shown in Figure 7. As we can see, some vectors are predominantly utilized by specific module or matrix types. For instance, vector #23 is heavily utilized in the formation of matrix A, while vector #29 is predominantly used in the formation of Query modules.

**Load balancing**   To demonstrate that the vector selection is free from load balancing issue, we present the vector usage in a Gemma-7B model trained on the MetaMathQA dataset, as shown in

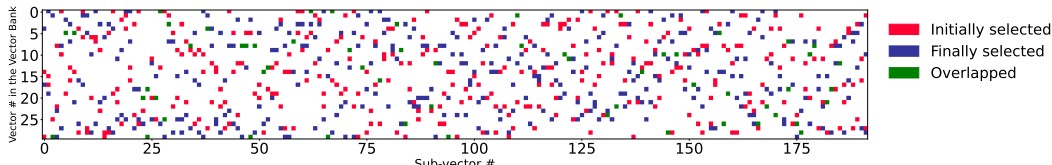

Figure 4: The x-axis represents the 192 sub-vectors formed by the vectors in the vector bank, while the y-axis represents the 30 vectors in the vector bank. The vectors initially selected by each sub-vector are shown in red, the vectors finally selected are shown in blue, and the overlapping vectors are shown in green.

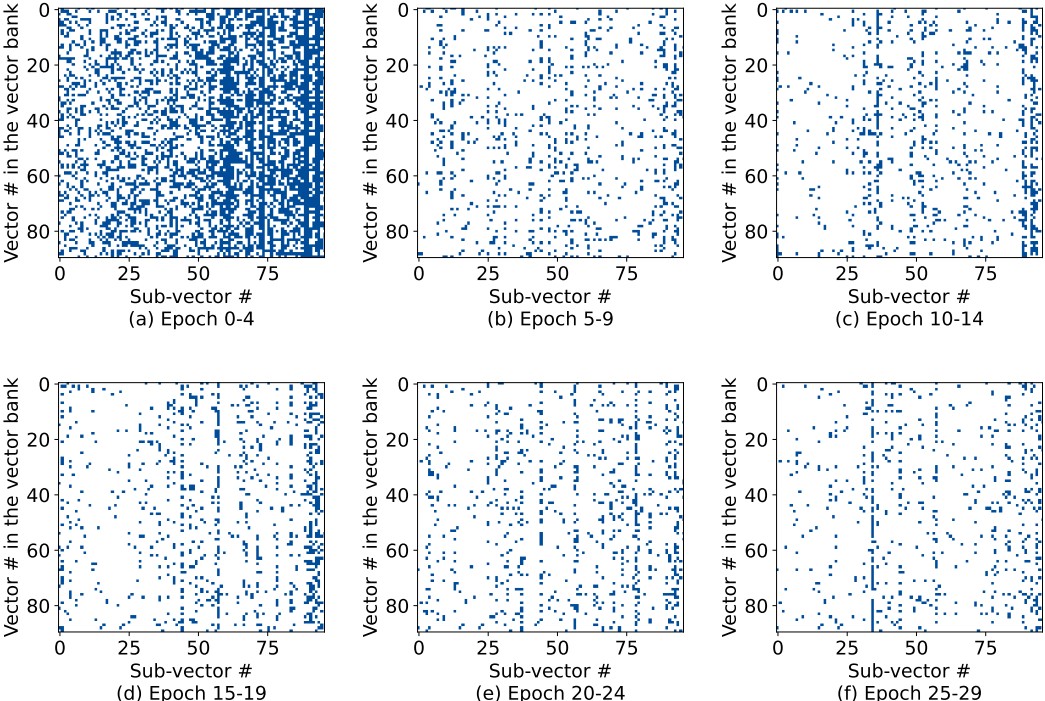

Figure 5: VB-LoRA's vector selection footprints during training. The x-axis represents the 96 sub-vectors formed by the vectors from a bank of 90 vectors, while the y-axis represents the indices of selected vectors from the bank. The blue blocks indicate the selection footprint during training.

Figure 8. The vector bank contains 2048 vectors. The distribution of vector usage follows a roughly normal distribution, with most vectors being selected between 40 to 55 times.

## A.3 Visualization of the Vector Bank and the Sub-vectors

Figure 9 illustrates the positioning of the sub-vectors along the edge of the simplex spanned by the vector bank. The vector bank is projected into a 2-D space using T-SNE [van der Maaten and Hinton, 2008] for visualization.

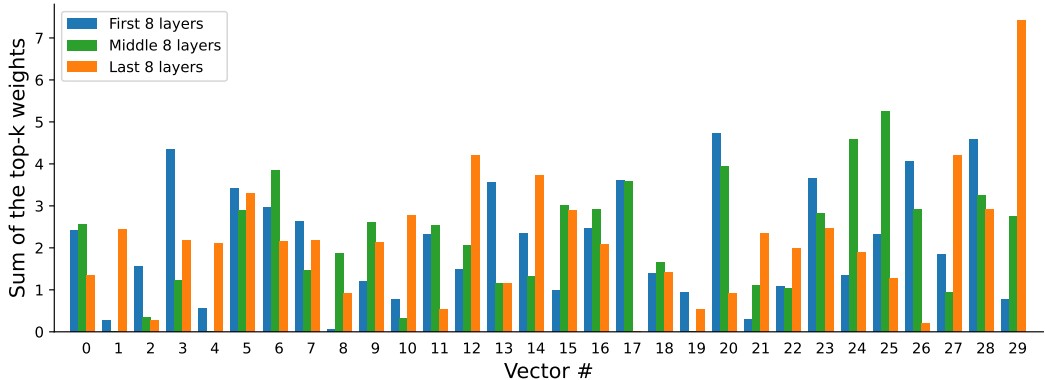

Figure 6: The sum of the top-$k$ weights for each vector, grouped by the first, middle, and last 8 layers. The vectors in $\mathcal{B}$ are sorted by their norms.

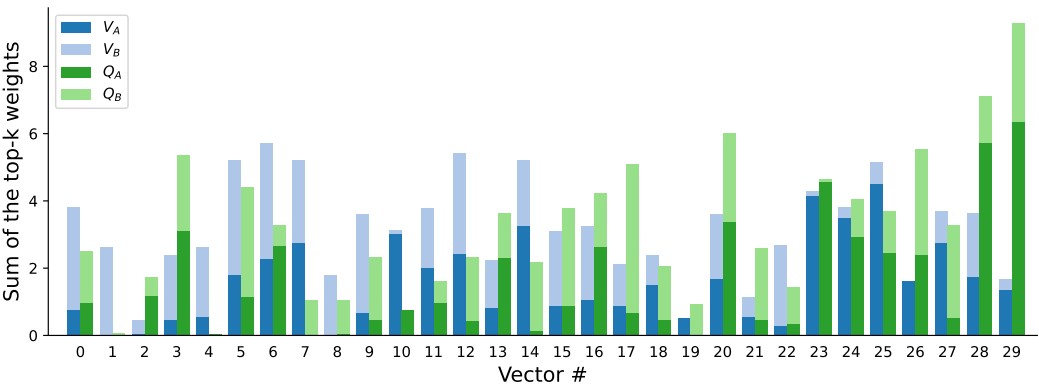

Figure 7: The sum of the top-$k$ weights for each vector, grouped by query (Q) and value (V) modules, and matrices A and B. The vectors in $\mathcal{B}$ are sorted by their norms.

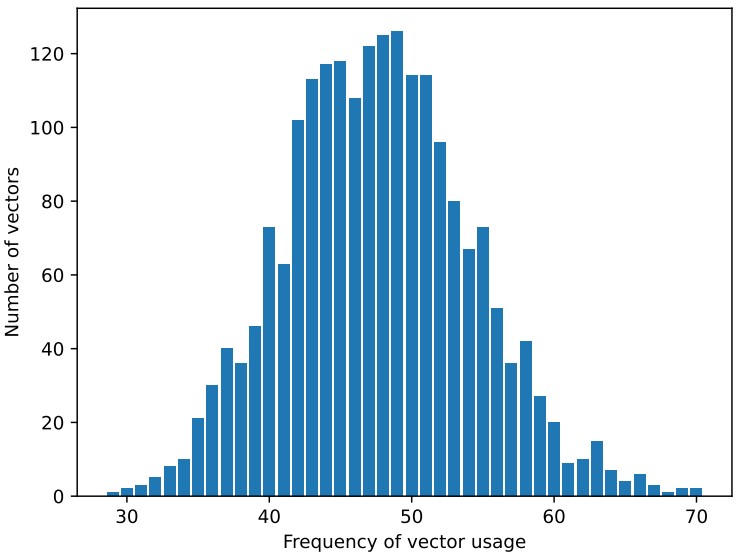

Figure 8: Histogram of vector usage frequency. The frequency ranges from 29 to 70, with most vectors being selected between 40 and 55 times. The distribution of vector usage follows an approximately normal pattern.

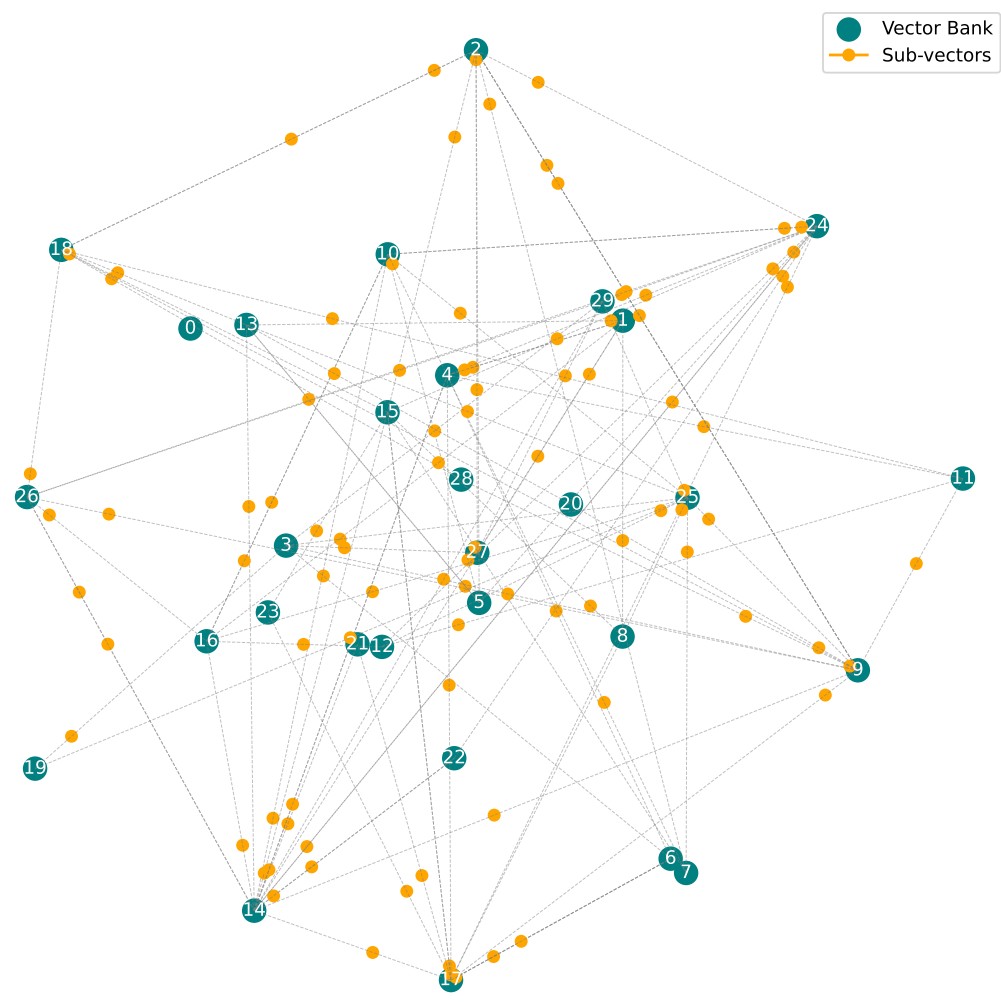

Figure 9: Visualization of the learned vector bank and sub-vectors.

## A.4 Instruction-tuning Examples

Tables 10 and 11 provide examples of the responses generated by the finetuned Llama2 13B model with LoRA and VB-LoRA.

Table 10: Example #1 of the responses generated by the finetuned Llama2 13B model with LoRA and VB-LoRA.

| | LoRA | VB-LoRA |
|---|---|---|
| | Draft a professional email seeking your supervisor's feedback on the 'Quarterly Financial Report' you prepared. Ask specifically about the data analysis, presentation style, and the clarity of conclusions drawn. Keep the email short and to the point. | |
| ANSWER | Dear [Supervisor's Name], I am writing to request your feedback on the quarterly financial report I prepared. I would appreciate your input on the data analysis, presentation style, and the clarity of the conclusions drawn. I have attached the report for your review. Please let me know if you require any additional information. Thank you for taking the time to review my work. I am eager to receive your feedback and learn from your insights. | Dear Supervisor, I am writing to request your feedback on the 'Quarterly Financial Report' I prepared. Specifically, I would like to know your thoughts on the data analysis, presentation style, and the clarity of conclusions drawn. As a reminder, I used the most recent data available to analyze the company's financial performance and provide insights into its overall health. I used charts and graphs to present the data in an easy-to-understand manner. Finally, I summarized the key findings and provided recommendations for future action. I look forward to your feedback and appreciate your time and consideration. Thank you. Your Name |
| REVIEW | The assistant's response is very professional, concise, and to the point. It clearly communicates the user's request for feedback on specific aspects of the report, namely the data analysis, presentation style, and clarity of conclusions. The assistant also shows respect for the supervisor's time and expresses a willingness to learn from their feedback. The response is relevant, accurate, and detailed enough for the context. However, it lacks a bit of creativity, but given the professional context, this is not a major issue. | The assistant's response is highly relevant, accurate, and helpful. It provides a professional and concise email draft that meets the user's requirements. The assistant has included all the necessary elements such as the specific areas where feedback is needed (data analysis, presentation style, and clarity of conclusions), a brief description of the work done, and a polite request for the supervisor's time and feedback. The response is also creative and detailed, providing a clear context for the supervisor. Therefore, I would rate it highly. |
| SCORE | 9 | **10** |

Table 11: Example #2 of the responses generated by the finetuned Llama2 13B model with LoRA and VB-LoRA. Parts of answers are omitted due to their length.

| | LoRA | VB-LoRA |
|---|---|---|
| | Compose an engaging travel blog post about a recent trip to Hawaii, highlighting cultural experiences and must-see attractions. | |
| ANSWER | Sure! Here is a travel blog post about a recent trip to Hawaii:
Hawaii is a paradise on earth, with its beautiful beaches, lush green landscapes, and warm tropical weather. My recent trip to the islands was filled with unforgettable experiences and breathtaking sights. The first stop on my itinerary was the city of Honolulu, the capital of Hawaii. I explored the historic landmarks, stunning beaches, and world-renowned cuisine. The Pearl Harbor Memorial was a must-visit, as it commemorates the tragic attack that occurred on December 7, 1941. The USS Arizona Memorial is located above the sunken battleship and is a solemn reminder of the sacrifices made during the war.
From Honolulu, I ventured to the island of Oahu, where I spent a few days relaxing on the beautiful Waikiki Beach. The beach is lined with palm trees and crystal-clear waters, and is the perfect spot for swimming, sunbathing, and snorkeling.After exploring the island, I ventured to Maui, known as the "Valley Isle." Maui is home to some of the most beautiful beaches in Hawaii, including the world-famous Ka'anapali Beach. I spent the day soaking up the sun and enjoying the warm, turquoise waters.
In conclusion, my trip to Hawaii was filled with unforgettable experiences and breathtaking sights. The islands are a paradise on earth, with something for everyone. Whether you're looking for history, culture, or natural beauty, Hawaii has it all. I can't wait to return and explore more of these beautiful islands. | Dear reader, I recently had the opportunity to travel to the beautiful island of Hawaii, and I am thrilled to share my experience with you. This trip was full of cultural immersion, adventure, and breathtaking scenery. Here is a glimpse of my journey:
On the first day, I landed in Honolulu, the capital city of Hawaii. I was immediately struck by the lush greenery and tropical weather. I spent the day exploring the city, visiting famous landmarks such as Pearl Harbor and the USS Arizona Memorial. The USS Arizona Memorial is a somber yet powerful tribute to the brave soldiers who lost their lives in the attack on Pearl Harbor.
I spent the next few days relaxing on the beautiful beaches of Maui. I went snorkeling in the crystal-clear waters, and saw tropical fish, turtles, and even a few sharks. The snorkeling was truly an unforgettable experience.
One of the highlights of my trip was a visit to the Polynesian Cultural Center. This is an interactive museum that showcases the culture and traditions of Polynesian islands. I got to try traditional dances, make fire, and even learn how to play the ukulele.
In conclusion, my trip to Hawaii was truly unforgettable. I experienced cultural immersion, adventure, and breathtaking scenery. If you're looking for a trip of a lifetime, I highly recommend visiting this beautiful island paradise. |
| REVIEW | The assistant's response is a well-written and engaging travel blog post about a recent trip to Hawaii. It provides a detailed itinerary of the trip, highlighting the cultural experiences and must-see attractions in each location. The assistant also does a good job of describing the natural beauty and unique features of each island, which adds depth and interest to the post. The assistant's response is relevant, accurate, and creative, making it a high-quality response to the user's request. However, it could have included more about the cultural experiences, such as local food, music, or traditions. | The assistant's response is highly detailed, engaging, and relevant to the user's request. It provides a comprehensive overview of a trip to Hawaii, highlighting cultural experiences and must-see attractions. The assistant's use of descriptive language helps to paint a vivid picture of the experiences, making the blog post more engaging for readers. The assistant also provides a personal touch by sharing their own experiences and impressions, which adds depth to the response. The assistant's response is accurate, as it mentions real places and experiences in Hawaii. Overall, the assistant's response is highly creative and provides a high level of detail, making it an excellent travel blog post. |
| SCORE | 8.5 | **10** |

