# OpenReview forum: "VB-LoRA: Extreme Parameter Efficient Fine-Tuning with Vector Banks"
_NeurIPS.cc/2024/Conference — NeurIPS 2024 poster_

### Official Review · Reviewer_o6ri · 2024-07-03

**Soundness:** 4
**Presentation:** 3
**Contribution:** 4
**Rating:** 8
**Confidence:** 5

**Summary:**

This paper proposes an approach to decrease the number of parameters in LoRA by introducing a collection (aka "vector bank") of sub-vectors and composing Lora matrices across modules/layers using this collection. Each sub-vector within a given LoRA module/layer is then formed as a linear interpolation of sub-vectors from the vector bank, where the coefficients are obtained by selecting top-k "basis vectors" and then applying a softmax. When applied to Roberta-large, the approach yields improvements over LoRA on the GLUE benchmark with only a tenth of the free parameters. When applied GPT-2 large , it obtains similar or slightly better performance than LoRA with around a sixth of the parameters, on the E2E task. The paper reports ablation studies showing the effect of the vector selection strategy (i.e. alternatives to top-k) and the sub-vector length.

**Strengths:**

* Proposes a novel approach to reduce the parameter count in LoRA by sharing parameters across modules and layers using a sub-vector approach. (Originality)
* The use of sub-vector style decomposition is novel in the context of LoRA and could be applied in other PEFT settings (Significance).
* Demonstrates that the approach can yield improvements over LoRA using Roberta or GPT-2 with far fewer parameters (Quality)

**Weaknesses:**

* The paper does not discuss if the proposed factorization approach (i.e. using subvectors) can be performed efficiently on accelerators such as GPUs. Also it would be good to know what is the overhead resulting from this subvector decomposition.
* It would be useful to report the performance of an alternative PEFT scheme such as prefix tuning (or prefix tuning) or adapters which have a similar number of parameters as the proposed approach.

**Questions:**

* It would be useful to discuss this paper in related work: https://openreview.net/pdf?id=qKQu1ZcJjD : The idea is to initialize matrices in an LLM using pre-trained submatrix components, which bears resemblance to the subvector decomposition.

Typos:
L101:
* parameters -> parameter
L192
* numebr -> number
L314
* frozen -> frozen

**Limitations:**

Yes

---

> ### Author Rebuttal · Authors · 2024-08-07
>
> Dear Reviewer o6ri,
>
> **1. Weakness #1 – GPU acceleration and computation overhead**
>
> VB-LoRA’s implementation is straightforward, and the proposed factorization approach is also simple to implement in modern deep learning frameworks such as PyTorch, allowing us to fully leverage GPU acceleration. However, the use of subvector decomposition does introduce some computational overhead. Despite this, the reduced number of trainable parameters in VB-LoRA results in lower memory consumption. As shown in Table 8, we compared the training time and memory usage of LoRA and VB-LoRA. VB-LoRA requires approximately 15%-20% more training time than LoRA, while using less memory. It's important to note that this additional overhead is limited to the training phase and does not affect inference, as both LoRA and VB-LoRA merge their parameters back into the original model parameters during this stage.
>
> **2. Weakness #2 – comparing with other PEFT methods**
>
> Thank you for your suggestion! We will revise the manuscript to include additional baseline comparisons. Below, we compare our method with inserted adapters and BitFit. Our results show that our method outperforms other PEFT methods while using an order of magnitude fewer parameters.
>
> |               | Method  | # Params | SST-2 | MRPC | CoLA | QNLI | RTE  | STSB | Avg. |
> |---------------|---------|----------|-------|------|------|------|------|------|------|
> | Roberta-base  | VB-LoRA | **0.027M**   | **95.0**  | 89.7 | **64.3** | 92.3 | **82.3** | 90.7 | **85.7** |
> |               | Adpt_D [1] | 0.3M     | 94.2  | 88.5 | 60.8 | **93.1** | 71.5 | 89.7 | 83.0 |
> |               | BitFit [4]  | 0.1M     | 93.7  | **92.7** | 62.0 | 91.8 | 81.5 | **90.8** | 85.4 |
> | Roberta-large | VB-LoRA | **0.033M**   | 96.3  | **91.9** | **69.3** | 94.4 | **87.4** | 91.8 | **88.5** |
> |               | Adpt_P [2] | 0.8M     | **96.6**  | 89.7 | 67.8 | **94.8** | 80.1 | **91.9** | 86.8 |
> |               | Adpt_H [3] | 0.8M     | 96.3  | 87.7 | 66.3 | 94.7 | 72.9 | 91.5 | 84.9 |
>
>
> 1. Adpt_D: Andreas Rücklé, Gregor Geigle, Max Glockner, Tilman Beck, Jonas Pfeiffer, Nils Reimers, and Iryna Gurevych. AdapterDrop: On the efficiency of adapters in transformers. In Proceedings of the 2021 Conference on Empirical Methods in Natural Language Processing, pp. 7930–7946.
> 2. Adpt_P: Jonas Pfeiffer, Aishwarya Kamath, Andreas Rücklé, Kyunghyun Cho, and Iryna Gurevych. AdapterFusion: Non-destructive task composition for transfer learning. In Proceedings of the 16th Conference of the European Chapter of the Association for Computational Linguistics: Main Volume, pp. 487–503.
> 3. Adpt_H: Neil Houlsby, Andrei Giurgiu, Stanislaw Jastrzebski, Bruna Morrone, Quentin de Laroussilhe, Andrea Gesmundo, Mona Attariyan, and Sylvain Gelly. Parameter-efficient transfer learning for nlp, 2019.
> 4. BitFit: Elad Ben Zaken, Shauli Ravfogel, and Yoav Goldberg. Bitfit: Simple parameter-efficient fine-tuning for transformer-based masked language-models, 2022.
>
>
> **3. Question #1 – related work “Deep Fusion”**
>
> Thank you for bringing Deep Fusion [1] to our attention! This work provides an efficient method for network training under the network growth mechanism by leveraging pre-trained initializations of smaller networks. It shares a similarity with our approach in that both methods induce structure on the parameters of the original transformer model for improved efficiency. Additionally, initialization can be seen as a way to share computation and reduce costs between training smaller and larger models. We will include a discussion of this work in the related work section of our manuscript.
>
> 1. Mazzawi et al., Deep Fusion: Efficient Network Training via Pre-trained Initializations, Forty-first International Conference on Machine Learning, 2024

---

> > ### Comment · Reviewer_o6ri · 2024-08-13
> >
> > Thanks to the authors for sharing the newer results and adding clarifications.

---

### Official Review · Reviewer_eNJC · 2024-07-09

**Soundness:** 2
**Presentation:** 2
**Contribution:** 2
**Rating:** 5
**Confidence:** 5

**Summary:**

The authors propose an extremely parameter-efficient methods, VB-LoRA, for finetuning an LLM. Specifically, VB-LoRA has a shared vector bank (similar to a codebook), the adapter parameters (A and B) of the linear layers in an LLM are constructed from this bank by selecting the most effective bank vectors. Due to the shared mechanism, the number of trainable parameters are very little, much fewer than the strong baselines, like LoRA, VeRA and Tied-LoRA.

Tha authors also validate VB-LoRA on three benchmarks (GLUE, E2E benchmark and MT-Bench) on 6 LLMs from three model families (RoBERTa, GPT-2 and Llama2). The results show that VB-LoRA's results are better or comparable to above-mentioned baselines while requiring much fewer number of trainable parameters.

**Strengths:**

1. The proposed method, VB-LoRA, is effective, intuitive and easily applied.
2. The number of trainable parameters are very limited, mostly less than 0.1M.

**Weaknesses:**

1. The paper is not easy to follow, especially Section 3 for the proposed method. VB-LoRA is basically an algorithm inspired by Mixture-of Expert or codebook, and is used for PEFT here. However, the authors seem to intend to make it complexer by relating VB-LoRA to other theoretical works (L137-143, L157-162), which would be better by putting in the related works. Such writing style easily distracts the reader's attention.

2. Some experimental setting is weird, not in line with previous work.
- In Table 1, two high-resource tasks of GLUE, i.e. MNLI and QQP, are deleted. The authors claim that the experiments are conducted on  a server equipped with 8 A100 GPUs (L215), which means the computation resource should not be a problem. Why are these two tasks are discarded?
- All ablation studies are conducted on a very small task, CoLA, whose results have large variance. Normally, 24GB GPU memory and less than 1 hour are enough for finetuning RoBERTa on CoLA. Due to the small number of samples in CoLA, its results have relatively larger variance. Since the authors have much more resource, why choosing this task for ablation study?

3. VB-LoRA has a potential drawback that is inherent in its design, i.e. unscalibility. One big problem for MoE is that not all expert embeddings can be well trained. We normally include a balance loss and use a small number of experts to avoid this problem. For VB-LoRA, could you offer this ablation study? I.e. increasing the number and length of the bank vectors.

4. VB-LoRA doesn't consistently outperform baselines, or the gap is very narrow. I would be interested in more benckmarks on larger LLMs, like commonsen/arithmetic reseasoning tasks.

**Questions:**

1. In algorithm 1, if you implement delta_W in this way, it will comsume more GPU memory during finetuning than LoRA. For implementing LoRA, the forward pass is z = Ax, H = Bz. It means we don't calculate delta_W explicitely. If you calculate delta_W, it will be stored in the memory for gradient computation, which enlarges the memoy for activation.

2. Check other questions in Weakness.

**Limitations:**

The limitation discussion is not thorough.

---

> ### Author Rebuttal · Authors · 2024-08-07
>
> Dear Reviewer eNJC,
>
> **1. Weakness #2 – why MNLI and QQP are discarded?**
>
> We adhered to the experimental settings established by our baseline, VeRA, which did not include MNLI and QQP. Despite having access to computational resources, it is a shared resource in university. We chose to focus on Natural Language Generation (NLG) and instruction tuning tasks and evaluation on GPT-2 and Llama-2 models, which we believe is of more interest to the research community.
>
> **2. Weakness #2 – why choose CoLA for ablation study?**
>
> It is a typical practice to choose smaller sized benchmarks for ablation study, since running small benchmarks across different experimental settings can still be computationally intensive and time-consuming.  We chose CoLA for the ablation study due to its relevance and consistency with the prior works, e.g., VeRA. Each experiment on the GLUE benchmark was run five times, and the results are reported as the median of these runs to mitigate variance.
>
> **3. Weakness #3 – scalability of VB-LoRA to larger models**
>
> Note that there are two distinct differences in the routing module between VB-LoRA and MoE. First, the expert network in VB-LoRA is simply a vector in the vector bank. Second, the gating network does not take any model input; instead, it is freely updated by gradients during fine-tuning. In our experiments (e.g., Figure 3, and Figure 5 in the appendix), we observed that even without noisy logits, our method does not suffer from the load balancing issue seen in MoE.
>
> In terms of scalability, we have conducted experiments with a range of model sizes, from Robota 125M to Llama 13B, and varied the number and length of vector banks from 30 to 2048 vectors and 128 to 1024 dimensions, respectively. Our method demonstrated consistent performance across these variations.
>
> To further demonstrate that our approach is free from load balancing issues that hurt scalability, we present the vector usage in a Gemma-7B model trained on the MetaMathQA dataset, as shown in Figure 2 of the attached PDF file. The vector bank contains 2048 vectors. The distribution of vector usage follows a roughly normal distribution, with most vectors being selected between 40 to 55 times.
>
> **4. Weakness #4 – performance of VB-LoRA**
>
> VB-LoRA's primary focus is on parameter efficiency rather than predictive performance. Despite the extremely high parameter efficiency, our method still delivers accuracies comparable to or sometimes surpassing baseline models.
>
> We appreciate the reviewer’s suggestion to evaluate our method on larger language models, particularly in commonsense and arithmetic reasoning tasks. In response, we have fine-tuned the Mistral-7B and Gemma-7B models on the MetaMathQA dataset and evaluated them on the GSM8K and MATH datasets. We compared our method with LoRA and the concurrent work LoRA-XS [1]. Results show that VB-LoRA outperforms all baselines on the GSM8K dataset, with Mistral-7B utilizing only 0.4% of the parameters compared to LoRA, and Gemma-7B using just 0.3%. Compared to LoRA-XS, our method achieved better results on both evaluation datasets while using only 70% of the parameters with Mistral-7B and 83% with Gemma-7B.
>
> | Model      | Method         | # Params | GSM8K | MATH  |
> |------------|----------------|----------|-------|-------|
> | Mistral-7B | Full FT        | 7242M    | 67.02 | 18.60 |
> |            | LoRA (r=64)    | 168M     | 67.70 | **19.68** |
> |            | LoRA-XS (r=64) | 0.92M    | 68.01 | 17.86 |
> |            | **VB-LoRA** | **0.65M**    | **69.22** | 17.90 |
> | Gemma-7B   | Full FT        | 8538M    | 71.34 | 22.74 |
> |            | LoRA (r=64)    | 200M     | 74.90 | **31.28** |
> |            | LoRA-XS (r=64) | 0.80M    | 74.22 | 27.62 |
> |            | **VB-LoRA** | **0.67M**    | **75.96** | 28.90 |
>
> 1. Bałazy, Klaudia, et al. "LoRA-XS: Low-Rank Adaptation with Extremely Small Number of Parameters." arXiv preprint arXiv:2405.17604 (2024).
>
>
> **5. Question #1 – memory consumption**
>
> We appreciate your insight into the memory consumption concerns. Indeed, the method you mentioned for reducing memory usage was considered and has been implemented already in our submitted supplementary source code, as it integrates seamlessly with our approach. Algorithm 1 serves as a high-level pseudocode to illustrate the formulation of matrices A and B, intentionally omitting the input x for simplicity. Thus, it does not explicitly showcase the memory-saving technique you referenced. We will revise Algorithm 1 as suggested and make it consistent to our implementation.

---

> > ### Comment · Reviewer_eNJC · 2024-08-13
> >
> > Thank you for the clarification and etra eperiments. I still have one more question for W3:
> >
> > Why don't you show the scalability results here? In stead, you only simply mentioned "we have conducted experiments with a range of model sizes, from Robota 125M to Llama 13B, and varied the number and length of vector banks from 30 to 2048 vectors and 128 to 1024 dimensions, respectively. Our method demonstrated consistent performance across these variations."

---

> > > ### Author Response · Authors · 2024-08-13
> > >
> > > Thanks for the follow-up question. We did the ablation study as you suggested, i.e., increasing the number and length of the bank vectors, to demonstrate the scalability. The table below shows the results on COLA (median over 5 runs, with standard deviations in parentheses).
> > >
> > > As we can see, as the bank size (90->8192) and vector length (256->1024) increase, the performance of VB-LoRA is relatively stable. Once the number of trainable parameters is very large, the performance degrades a little bit, showing the sign of overfitting. This result further illustrates the scalability of our method.
> > >
> > > We will include this table in the revised paper.
> > >
> > > | Bank size | Vector length | # Params (M) | COLA  |
> > > |-----------|---------------|--------------|-------|
> > > | 90        | 256           | 0.03         | 69.3 (1.5) |
> > > | 1024      | 256           | 0.28         | 69.2 (1.2) |
> > > | 8192      | 256           | 2.12         | 69.4 (1.3) |
> > > | 8192      | 1024          | 8.39         | 68.8 (0.7) |

---

> > > > ### Comment · Reviewer_eNJC · 2024-08-14
> > > >
> > > > Thank you for the new results. I'm willing to raise my score to 5, because some of my following concerns are not fully addressed:
> > > >
> > > > 1. **CoLA for ablation study**. I really suggest to use a more recent dataset (like math related or commonsense related datasets) and recent LLMs (Llama2-7B) to reproduce the ablation study. CoLA only contains 8.5K samples with short sentences. As your new results shown above, the variance is about 1 (much larger than the improvement), which makes the observation less convincing.
> > > >
> > > > 2. **Scalability**: As your new results shown above, VB-LoRA's scalability is not good. When increasing the bank size or the vector length, the performance doesn't really change. This may hint that VB-LoRA is not easy to be applied to knowledge-intensive tasks. Please include a related discussion in the limitation section, which is very important for your followers.

---

### Official Review · Reviewer_wW2k · 2024-07-10

**Soundness:** 3
**Presentation:** 4
**Contribution:** 3
**Rating:** 7
**Confidence:** 4

**Summary:**

This paper explores parameter-efficient fine-tuning (PEFT) methods in the context of further reducing the number of fine-tunable parameters, even to the extreme. The main idea is to reduce the number of parameter within LoRA modules as much as possible while maintaining or even improving fine-tuning performance. This line of work is a sub-area of PEFT research, where similar ideas have been explored in the past (e.g., the work of Karimi Mahabadi et al. on Compacter architectures, or a very recent work concurrent to this one: LoRA-XS).

To this end, the authors propose VB-LoRA which aims to use globally shared vector banks comprising 'constituent' sub-vectors that can be recombined into local 'delta' parameters (of the \Delta W matrix) that modify/adapt the weights of the original model. The VB-LoRA is then evaluated on GLUE with RoBERTa (for NLU), E2E with GPT-2 (for generation), and Llama 2 on MT-Bench (for instruction tuning) to show its benefits in terms of paramater efficiency and performance.

**Strengths:**

- The work is well situated and shows excellent awareness of other work in this subarea which also partially inspired the proposed VB-LoRA method. The chosen baseline models are largely adequate and the only baseline missing (to the best of my knowledge) is LoRA-XS (which is concurrent work, so it couldn't have been included anyhow).

- The method is well motivated from a conceptual level and it is also well described formally, in a succinct and convincing manner. The idea does resemble matrix factorisation and tensor product-inspired methods a lot, so I would like to see additional links to previous literature here.

- I appreciate the fact that the authors aimed to provide a comprehensive evaluation over NLU, NLG as well as instruction-tuned models. However, this approach has traded some depth of evaluation for its breadth (and the evaluation is not entirely adequate and comprehensive - see under "Weaknesses" later).

- A careful derivation related to the number of fine-tuned parameters across different methods is a very useful addition to the paper, given the fact that the parameter efficiency-performance balance is the main topic of the paper.

- The paper is very clear and well written in general.

**Weaknesses:**

- Evaluation is not comprehensive:
1) When it comes to instruction-tuned models, the main findings are based on a single model from a single family (Llama 2), fine-tuned on a single instruction tuning dataset and evaluated on a single evaluation dataset. This is simply enough to move from anecdotal to more generalisable findings, and experiments with (i) additional models, (ii) more instruction tuning datasets (e.g., Flan-v2, Tulu v2, there are more), and (iii) additional evaluation sets (e.g., MMLU, GSM, BBH, HumanEval) are required.
2) The same holds for NLU and NLG experiments. NLU is evaluated only on GLUE (where it's widely established that performance on GLUE is already saturated and considered 'superhuman'). For NLG, only experiments with a single model on a single benchmark are run. As I mentioned above, the experiments are broad, but stay quite shallow, and this has to change in the revised version.

- The paper is very quantitatively driven, but it doesn't delve into providing clarifications and explanations on why certain quantitative phenomena/findings are observed:
1) What is the intuition behind VB-LoRA outperforming LoRA? How can we explain this? What makes VB-LoRA even more suitable performance-wise? Isn't it counter-intuitive that an extremely efficient model can outperform here? Are the tasks then too simple (with very low intrinsic dimensionality) as is the case with some of the standard GLUE tasks?
2) How can one fine-tune the optimal VB size (b) and dimensionality of subvectors within the VB (Table 5) - is this task specific or model specific? What would happen if one increases parameter budget a bit - can we expect even better performance or not? In what cases?
3) Are there any sub-vectors that get selected more often than some others? Why?

Overall, while I'm quite happy regarding technical aspects of the work, the paper can be much improved in terms of evaluation/experiments and discussion of the key findings (some error analysis would be useful as well).

**Questions:**

- Lines 180-182. "It’s worth mentioning that adding the multi-index information to the vector selection mechanism can make the TKAM model structure-aware, potentially yielding additional benefits." This is unclear to me and warrants further clarification/discussion. Could you write a short paragraph on how the multi-index information would be added and what type of structure would then TKAM learn (become aware of)?

- Additional discussion on the similarities and differences between TKAM and standard (sparse and soft) MoE-s is required. Are there any MoE-style paradigms (which incur a higher parameter cost) that could have been used instead of TKAM? In general, the work doesn't really examine the choice of TKAM as a go-to routing method for the selection of sub-vectors.

- Can the authors elaborate more on the similarities and differences between their work and the Compacter work (beyond the fact that Compacter focuses on bottleneck adapters while VB-LoRA focuses on LoRA)? Conceptually, the papers operate in the same space with similar ideas.

**Limitations:**

There are limitations to this work that should be further elaborated (see also under 'Weaknesses'). The paper didn't explore the whole space of options when it comes to vector selection, VB size, VB content interpretation, and task dependency of findings, among other things. The work also stays within the realm of monolingual (i.e., English-only), mono-modal (i.e., text-only) and mono-PEFT (i.e., LoRA-only) contexts, and this should also be adequately signalled in the Limitations section.

---

> ### Author Rebuttal · Authors · 2024-08-07
>
> Dear Reviewer wW2k,
>
> **1.  Weakness #1 – evaluations on instruction tuning**
>
> Thank you for your suggestion. In response, we fine-tuned the Mistral-7B and Gemma-7B models on the MetaMathQA dataset and evaluated them on GSM8K and MATH datasets, and compared them with the suggested work LoRA-XS. Our experimental setup follows the LoRA-XS configuration. The results show that our method outperforms all baselines on GSM8K, with Mistral-7B utilizing only 0.4% of the parameters compared to LoRA, and Gemma-7B using just 0.3%. Compared with LoRA-XS, our method outperforms on both evaluation datasets while using 70% (Mistral-7B) and 83% (Gemma-7B) of parameters. More details can be found in the Global Rebuttal.
>
> | Model      | Method         | # Params | GSM8K | MATH  |
> |------------|----------------|----------|-------|-------|
> | Mistral-7B | Full FT        | 7242M    | 67.02 | 18.60 |
> |            | LoRA (r=64)    | 168M     | 67.70 | **19.68** |
> |            | LoRA-XS (r=64) | 0.92M    | 68.01 | 17.86 |
> |            | **VB-LoRA** | **0.65M**    | **69.22** | 17.90 |
> | Gemma-7B   | Full FT        | 8538M    | 71.34 | 22.74 |
> |            | LoRA (r=64)    | 200M     | 74.90 | **31.28** |
> |            | LoRA-XS (r=64) | 0.80M    | 74.22 | 27.62 |
> |            | **VB-LoRA** | **0.67M**    | **75.96** | 28.90 |
>
>
> **2. Weakness #1 – evaluations on NLU and NLG experiments**
>
> We chose the GLUE benchmark for evaluation to keep consistency with our baseline methods, LoRA and VeRA, both of which use GLUE in their experiments. Although performance on GLUE is considered saturated, we believe that using this benchmark does not undermine our evaluation, as our primary focus is on parameter efficiency rather than absolute performance metrics. Moreover, we have evaluated our method across a range of model sizes from 125M (Robota-base) to 13B (Llama 2). Overall, these experiments demonstrate the robustness and effectiveness of our method.
>
> **3. Weakness #2 – the intuition behind VB-LoRA**
>
> We agree that the performance of PEFT methods strongly depends on the task's intrinsic dimensionality [1]. In essence, It's not just the number of training parameters that matters, but also the way they are composed. While both methods use low-rank decomposition, VB-LoRA goes further by breaking layer and module boundaries, employing a less flexible yet expressive reparameterization based on a sparse convex combination of vectors. This approach introduces an inductive bias that can benefit tasks. For more complex tasks like mathematical reasoning, we've included additional evaluations to showcase the effectiveness of our method.
>
> 1. Aghajanyan, Armen, Luke Zettlemoyer, and Sonal Gupta. "Intrinsic dimensionality explains the effectiveness of language model fine-tuning." arXiv preprint arXiv:2012.13255 (2020).
>
> **4. Weakness #2 – the optimal VB size b and dimensionality of subvectors**
>
> Yes, the number of vectors (h) and the length of each vector (b) in the vector bank are task-specific hyperparameters that need to be tuned. Given a fixed budget (h x b), the performance is not highly sensitive to these parameters, as shown in Table 5.
>
> In general, increasing the parameter budget improves performance, as demonstrated in Figure 1. However, the extent of improvement also depends on other factors such as the model architecture and the task difficulty. Performance gains will plateau once these limitations are reached.
>
> **5. Weakness #2 – vector selection**
>
> In Figure 2 of the attached PDF, we present the vector usage for a Gemma-7B model trained on the MetaMathQA dataset. The distribution of vector usage follows an approximately normal pattern, with most vectors being selected between 40 to 55 times. However, some vectors are chosen more often, with some being selected up to 70 times.
>
> **6. Question #1 – adding the multi-index information to the vector selection mechanism can make the TKAM model structure-aware**
>
> We apologize for the confusion. Currently, VB-LoRA is not structure-aware, in the sense that the selection of vectors from the vector bank is not informed by factors such as module, layer, and matrix type. However, our framework can be easily extended to become structure-aware. One approach is to make the logits of vector selection conditional on embeddings of the layer, module, and matrix type, which can be implemented through a hypernetwork [1].
>
> 1. Mahabadi, Rabeeh Karimi, et al. "Parameter-efficient Multi-task Fine-tuning for Transformers via Shared Hypernetworks." Annual Meeting of the Association for Computational Linguistics, 2021.
>
> **7. Question #2 – TKAM and MoE**
>
> We chose TKAM for its simplicity and strong performance. In our ablation study (Section 4.4), TKAM outperforms other baselines like Gumbel-Softmax and Straight-Through Gumbel-Softmax. While alternatives like DSelect-k [1] and Expert Choice routing [2] are worth exploring, we focused on introducing the general idea in this paper.
>
> 1. Hazimeh, Hussein, et al. "Dselect-k: Differentiable selection in the mixture of experts with applications to multi-task learning." Advances in Neural Information Processing Systems, 2021.
> 2. Zhou, Yanqi, et al. "Mixture-of-experts with expert choice routing." Advances in Neural Information Processing Systems, 2022.
>
> **8. Question #3 – VB-LoRA and Compacter**
>
> Compacter and our work are conceptually similar, as both are PEFT methods that operate in an efficiently reparameterized space and reduce redundancies by globally sharing information. However, Compacter designates shared and adapter-specific parameters, while our approach introduces a vector bank with a learned sharing mechanism. This mechanism offers more flexibility by allowing dynamic, context-dependent utilization of shared parameters instead of static designation. We will include this discussion of the relationship between VB-LoRA and Compacter in the related work.
>
> **9. Limitations**
>
> We agree. We will include these points in the limitations section.

---

> > ### Comment · Reviewer_wW2k · 2024-08-13
> > **The response clarifies some of my concerns**
> >
> > I would like to thank the authors for the provided response and the additional results, and I'm happy to increase my score in light of the provided new evidence.

---

### Official Review · Reviewer_e7Nd · 2024-07-13

**Soundness:** 4
**Presentation:** 4
**Contribution:** 4
**Rating:** 8
**Confidence:** 4

**Summary:**

The authors present a modified version of LoRA called VB-LoRA which is a highly parameter efficient fine-tuning method. It uses a vector bank to represent the model parameters as a composition of vectors. This vector bank is then used to select top-k vectors using the top-k softmax function which are thereby pooled and arranged to form the A and B low-rank matrices of LoRA. The top-k admixture module function selected is a differentiable function so the whole model is trained end-to-end making it more efficient for a particular task. Ultimately the authors show that such as divide-and-share approach results in an orders of magnitude more parameter efficient technique.

**Strengths:**

1. The most significant contribution of this paper is to develop an LoRA-like adaptation technique that is orders of magnitude more parameter efficient than LoRA but at the same time maintaining comparable or better performance in all the tasks. VB-LoRA has the best average performance in all the tasks for every model.
2. The second most important aspect of VB-LoRA is that the number of parameters do not grow linearly with the model dimension (number of layers and model dimension).  Making the 'k' of top-k much smaller than This is highly valuable as the model sizes are getting increasingly bigger by the day.
3. The usage of differentiable top-k softmax (TKAM - mentioned in eqn. 1) is a great idea as it enables the paper to be trained fully for a specific task. Also dicing the vector to sub vectors of same size was great for sharing among all layers.
4. The authors have tried to go to the extreme in making the LoRA parameter count efficient by dividing even the low-rank vectors of LoRA and pooling them from a common vector bank.
5. The authors also provide adequate supplementary information such as the detailed hyper-parameters and the hardware used which is highly appreciated as it is highly valuable for anyone who wants to replicate these results.
5. The authors also provide the exact code as a supplementary material which is a huge plus, especially as the paper consists of several moving parts.
6. The paper is presented in a very high quality way using multiple different concepts and giving proper references wherever needed. This fact would be appreciated by the readers as the paper uses several different concepts to come up with their design such as sparse admixture models, top-k gating module, canonical polyadic decomposition, etc.

**Weaknesses:**

1. The parameter count of VB-LoRA is defined by: hb + 1.5LMr(d/b). The second term of this is still linearly dependent on the model dimensions (L and M). In the paper's experiments r<<b in which case the second is quite small and doesn't impact the growth in parameter by much. But there exist several adaptation tasks in which much larger ranks are needed. For example this paper (https://ieeexplore.ieee.org/abstract/document/10445894) shows that for domain adaptation large ranks (eg. 64) are much better than smaller ranks. In some of my other works for domain and language adaptation I have needed to use even larger ranks (200+) which are very much comparable to the value of b. In such cases, VB-LoRA's parameter requirement would be very much comparable to other methods.

2. Some parts of the paper are not entirely clear. For eg. in lines 166-169, the paper mentions that decomposed cumulative gradient parameters are more sensitive than the original model parameters during the training process. An intuition to why this is happening would be helpful.

3. In line 153, the paper mentions that by keeping k << h makes the learnt vectors highly specialized. However figure 3 shows that even with k's value as small as 2 and 3, the model updates almost all the vectors during the training process. Inherently this means that almost all the vectors have some activation and they are not very specialized.

4. Not a lot of details are provided for the virtual dimension 'b' and the results of table 5 are not explained.

5. nit: line 167: 'parameters'

**Questions:**

1. Can you provide an intuition to your finding why the cumulative gradient parameters are more sensitive than than the original model parameters and how this affects adaptation?
2. Have you experimented with larger ranks and how do they look in terms of performance and parameter count?
3. Have you tried this technique on other tasks than what's listed in the paper? Also have you tried it on other architectures?

**Limitations:**

1. In my view the authors have correctly mentioned that this paper has no larger societal impact beyond what LLMs may have.
2. Table-3 shows that the performance of LoRA may have improved with the changes in the GPT-4 model over time. But for the other tables the authors have used the old results for several competing models.
3. I agree with the rest of the answers to the NeurIPS paper checklist that the authors have provided.

---

> ### Author Rebuttal · Authors · 2024-08-07
>
> Dear Reviewer e7Nd,
>
> **1. Weakness #1, Question #2 – the parameter efficiency when rank is high**
>
> First, it’s important to note that rank may not be directly comparable across different methods. In many approaches, rank dictates the number of independent trainable parameters. However, in our method, the majority of the trainable parameters reside in the vector bank, allowing us to use a smaller rank compared to other methods. For instance, when training LLaMA2-13B for instruction tuning, VB-LoRA performs well with r=6, whereas LoRA requires r=64 and VeRA requires r=1024. New experiments on Mistral-7B and Gemma-7B also show that the rank in our method (rank=4) can be significantly lower than baseline methods (r=64), while achieving better performance. More details can be found in the Global Rebuttal.
>
>
> | Model      | Method         | # Params | GSM8K | MATH  |
> |------------|----------------|----------|-------|-------|
> | Mistral-7B | Full FT        | 7242M    | 67.02 | 18.60 |
> |            | LoRA (r=64)    | 168M     | 67.70 | **19.68** |
> |            | LoRA-XS [1] (r=64) | 0.92M    | 68.01 | 17.86 |
> |            | **VB-LoRA** (r=4) | **0.65M**    | **69.22** | 17.90 |
> | Gemma-7B   | Full FT        | 8538M    | 71.34 | 22.74 |
> |            | LoRA (r=64)    | 200M     | 74.90 | **31.28** |
> |            | LoRA-XS [1] (r=64) | 0.80M    | 74.22 | 27.62 |
> |            | **VB-LoRA** (r=4) | **0.67M**    | **75.96** | 28.90 |
>
> If high rank is needed, VB-LoRA can reduce the number of parameters more significantly compared to other PEFT methods due to the global sharing mechanism. In this case, the gap in parameter efficiency between LoRA and Full FT becomes smaller, whereas VB-LoRA can sustain a high rank while maintaining a compact vector bank.
>
> 1. Bałazy, Klaudia, et al. "LoRA-XS: Low-Rank Adaptation with Extremely Small Number of Parameters." arXiv preprint arXiv:2405.17604 (2024).
>
>
> **2. Weakness #2, Question #1 – decomposed cumulative gradient parameters are more sensitive than the original model parameters during the training process**
>
> The reason for this increased sensitivity is the decomposition process. Similar to hypernetwork, the parameters are generated by another parameterized model, instead of directly being updated through gradients. In our case, the cumulative gradient updates ΔW are decomposed into matrices A and B, and further into sub-vectors in VB-LoRA. The vectors in the vector bank and logits are the parameters directly updated through gradients. This can be related to the training instability issues observed in hypernetworks [1], where heuristics such as gradient clipping are typically used to manage these instabilities [2]. In TKAM, we found that adding noise to the logits [3] can lead to difficulties in training stability.
>
> 1. Ortiz, Jose Javier Gonzalez, John Guttag, and Adrian V. Dalca. "Magnitude Invariant Parametrizations Improve Hypernetwork Learning." The Twelfth International Conference on Learning Representations. 2024.
> 2. Ha, David, Andrew Dai, and Quoc V. Le. "Hypernetworks." arXiv preprint arXiv:1609.09106 (2016).
> 3. Shazeer, Noam, et al. "Outrageously large neural networks: The sparsely-gated mixture-of-experts layer." arXiv preprint arXiv:1701.06538 (2017).
>
> **3. Weakness #3 – vector specialization**
>
> Figure 3 shows the footprint of the entire training process. It is important to note that active exploration predominantly occurs in the early stages of training. As training progresses, each sub-vector starts to focus more on much fewer (specialized) vectors within the vector bank. To highlight this pattern, we have plotted the footprint at different training periods in Figure 1 in the attached pdf file. This visualization demonstrates that updates become progressively sparser in the later stages of training.
>
> **4. Weakness #4 – virtual dimension b**
>
> The number of vectors (h) and the length of each vector (also known as the virtual dimension b) in the vector bank are task-specific hyperparameters that need to be tuned. One constraint on b is that it needs to be a common factor of all hidden dimensions to ensure compatibility across the entire model, as the hidden dimensions of FFN layers may differ from those of attention layers.
>
> Table 5 shows that given a fixed budget (h×b), the performance is not highly sensitive to the exact values of these parameters. However, selecting a moderate b may help achieve higher performance. Given the budget of the vector bank, a larger b reduces the number of vectors in the vector bank, potentially making each vector less specialized. On the other hand, a smaller
> b increases the number of trainable parameters (as discussed in Sec. 3.4) and complicates the vector selection process.
>
> We will include this discussion in the paper.
>
> **5. Limitation #2 – evaluations with GPT-4**
>
> GPT-4 is only used for evaluating instruction tuning for Llama2. All other experiments do not rely on GPT-4, thus remaining consistent over time.

---

### Author Rebuttal · Authors · 2024-08-07

We thank all reviewers for their valuable opinions and comments. In response to the reviewers' requests, we have added additional mathematical reasoning experiments for the Mistral-7B and Gemma-7B models.

###  Mathematical Reasoning Experiments

We fine-tuned the Mistral-7B-v0.1 and Gemma-7B models on the MetaMathQA dataset and evaluated them on GSM8K and MATH datasets. We compared our results with the concurrent work LoRA-XS [1]. Our experimental setup follows the LoRA-XS configuration. We use a vector bank size of 2048 with b=256, and set the rank to 4. We use a batch size of 128 and train for 2 epochs. The warmup ratio is 0.02, and a cosine learning rate scheduler is used. The initial learning is set to 0.001 for the vector bank and 0.01 for the logits. The experiment is performed on A100 80GB GPUs.

**The results show that our method outperforms all baselines on GSM8K, with Mistral-7B utilizing only 0.4% of the parameters compared to LoRA, and Gemma-7B using just 0.3%. Compared with LoRA-XS, our method outperforms on both evaluation datasets while using 70% (Mistral-7B) and 83% (Gemma-7B) of LoRA-XS parameters.**

Table 1. Instruction tuning on GSM8K and MATH Benchmarks for Mistral-7B and Gemma-7B models.
| Model      | Method         | # Params | GSM8K | MATH  |
|------------|----------------|----------|-------|-------|
| Mistral-7B | Full FT        | 7242M    | 67.02 | 18.60 |
|            | LoRA (r=64)    | 168M     | 67.70 | **19.68** |
|            | LoRA-XS (r=64) | 0.92M    | 68.01 | 17.86 |
|            | VB-LoRA (Ours) | **0.65M**    | **69.22** | 17.90 |
| Gemma-7B   | Full FT        | 8538M    | 71.34 | 22.74 |
|            | LoRA (r=64)    | 200M     | 74.90 | **31.28** |
|            | LoRA-XS (r=64) | 0.80M    | 74.22 | 27.62 |
|            | VB-LoRA (Ours) | **0.67M**    | **75.96** | 28.90 |


1. Bałazy, Klaudia, et al. "LoRA-XS: Low-Rank Adaptation with Extremely Small Number of Parameters." arXiv preprint arXiv:2405.17604 (2024).


The attached PDF file contains Figure 1: VB-LoRA’s vector selection footprints during training, and Figure 2: Histogram of vector usage frequency.

---

### Decision · Program_Chairs · 2024-09-25

**Decision:**

Accept (poster)

**Comment:**

Reviewers consistently praise VB-LoRA's ability to achieve comparable or even better performance than strong baselines like LoRA and VeRA while using significantly fewer trainable parameters (often less than 0.1M). This is a major contribution in the growing field of PEFT. The paper provides a reasonably comprehensive evaluation across different tasks (NLU, NLG, Instruction Tuning) and model families (RoBERTa, GPT-2, Llama2). Furhter, the the core concept of a shared vector bank and composing adapter parameters from it is intuitive and easy to implement. The most critical reviewer, eNJC raised concerns about the scalability of VB-LoRA, particularly the potential for load balancing issues similar to MoE. The authors clarified the differences in the routing mechanism between VB-LoRA and MoE, arguing that VB-LoRA is less prone to load balancing problems. They also provided new experimental results showcasing a more balanced distribution of vector usage in a larger model (Gemma-7B).

I recommend accepting this paper. As the authors agreed to do, please expand the limitations section based on reviewer feedback. This should include acknowledging the limitations in the explored space of options (vector selection, VB size, etc.) and the focus on monolingual, mono-modal, and LoRA-only contexts.